# SoTA with Less: MCTS-Guided Sample Selection for Data-Efficient Visual Reasoning Self-Improvement

**Xiyao Wang[1,2†], Zhengyuan Yang[2], Chao Feng[3,4], Hongjin Lu[1]**
**Linjie Li[2], Chung-Ching Lin[2], Kevin Lin[2], Furong Huang[1,‡], Lijuan Wang[2,‡]**
[1]University of Maryland, College Park    [2]Microsoft
[3]University of Michigan    [4]Cornell University
[†]xywang@umd.edu    [‡]Equal advise

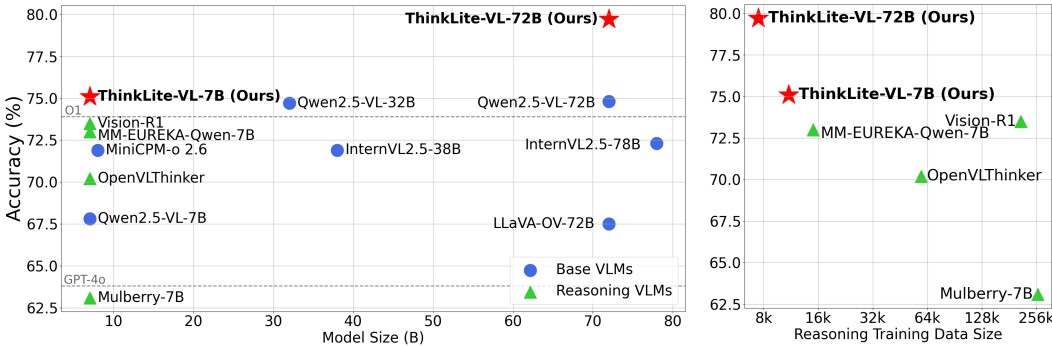

Figure 1: Recent "Reasoning VLMs" studies finetune "Base VLMs" with extra reasoning training data to improve visual reasoning. This paper presents a data-efficient self-improving method for better training reasoning VLMs. **(Left)** Comparison of VLMs with different parameter sizes on **MathVista**. Our model ThinkLite-VL-7B achieves the state-of-the-art (SoTA) accuracy of **75.1**, surpassing Qwen2.5-VL-72B-Instruct, GPT-4o, O1, and other 7B-level reasoning VLMs. ThinkLite-VL-72B further pushes this boundary to **79.7**. **(Right)** Comparison of the reasoning training data size used by 7B-level and 72B-level reasoning models. Our model achieves SoTA performance using only 11k data (7B) and 7.5k data (72B), and without any additional knowledge distillation.

## Abstract

We introduce **ThinkLite-VL**, a family of visual reasoning models that achieve state-of-the-art (SoTA) performance using an order of magnitude fewer training samples, relying purely on reinforcement fine-tuning (RFT) self-improvement without any knowledge distillation. Our central insight is that *sample difficulty* critically influences RFT effectiveness: appropriately challenging examples can drive substantial reasoning improvements, even in low-data regimes. However, quantifying sample difficulty in a reliable and scalable manner remains non-trivial. To address this, we repurpose Monte Carlo Tree Search (MCTS) to measure sample difficulty via the number of reasoning iterations a vision-language model (VLM) requires to solve each instance. This MCTS-based selection procedure identifies samples that induce deeper reasoning while remaining solvable, allowing us to filter a high-quality subset from 70k open-source examples spanning math, natural image understanding, and chart comprehension. Using this approach, we select just 11k challenging samples for RFT on Qwen2.5-VL-7B-Instruct and 7.5k samples for Qwen2.5-VL-72B-Instruct. The resulting models, ThinkLite-VL-7B and ThinkLite-VL-72B, significantly outperform their respective base models across eight visual reasoning benchmarks. In particular, ThinkLite-VL-7B improves the average

performance of Qwen2.5-VL-7B-Instruct by 7% and surpasses all existing 7B-level models, as well as much larger models such as GPT-4o, O1 and Qwen2.5-VL-72B, achieving a new SoTA score of **75.1** on MathVista. ThinkLite-VL-72B further advances the SoTA frontier, achieving an accuracy of **79.7** on MathVista and an average benchmark improvement of **4.42** over the open-source SOTA. These results demonstrate that MCTS-guided difficulty filtering provides a scalable and effective path toward data-efficient self-improvement in multimodal reasoning. Our code, data, and model are available at https://github.com/si0wang/ThinkLite-VL.

# 1 Introduction

Large language models (LLMs) have demonstrated strong capabilities in solving complex reasoning tasks—such as mathematics and coding—by leveraging chain-of-thought prompting and reflection mechanisms [24, 36]. Recent work [16] highlights the critical role of reinforcement fine-tuning (RFT) in further enhancing reasoning performance. Remarkably, these improvements can be achieved purely via RFT, even without post-training supervised fine-tuning (SFT).

However, despite the success of RFT in LLMs, its impact on vision-language models (VLMs) has been less pronounced. A likely cause is the inherent modality gap: VLMs are pretrained on text-heavy objectives, while post-training tasks demand multimodal reasoning. Recent efforts [22, 12, 55, 84] have addressed this by incorporating knowledge distillation and supervised format alignment before RFT. While effective, these pipelines are cumbersome, and fundamentally limit the capacity for models to improve via self-training alone.

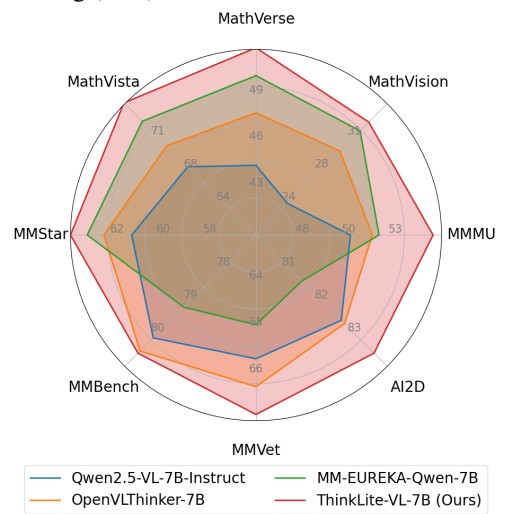

Figure 2: Performance comparison on 8 visual benchmarks. Our model significantly outperforms Qwen2.5-VL-7B and other reasoning models.

In this work, we demonstrate that high-quality and appropriately *challenging* training samples alone are sufficient to enable self-improvement in VLMs via RFT—without any knowledge distillation. When the training data matches the base model's capability level, RFT can explore informative rollouts by itself and substantially elevate multimodal reasoning ability. Based on this insight, we introduce **ThinkLite-VL**, a family of data-efficient reasoning VLMs trained via RFT on a small subset of difficulty-curated examples.

The key to ThinkLite-VL's performance lies in effective sample selection. We propose to repurpose Monte Carlo Tree Search (MCTS)—a classic inference-time search algorithm—to estimate the difficulty of each training instance. Specifically, we define difficulty as the number of MCTS reasoning iterations a VLM requires to solve a task. This search-based signal tightly correlates with sample difficulty and naturally identifies examples that promote deeper reasoning during training.

Our pipeline begins with 70k open-source samples spanning three core domains: mathematical reasoning, natural image understanding, and chart interpretation. For each example, we simulate an MCTS-based inference trace using the base VLM, and rank samples by the number of reasoning steps required to reach a correct solution. From this pool, we extract two difficulty-filtered subsets: 11k samples for Qwen2.5-VL-7B-Instruct and 7.5k samples for Qwen2.5-VL-72B-Instruct. We then apply RFT directly on these subsets—no supervised fine-tuning or distillation required.

We evaluate our resulting models, ThinkLite-VL-7B and ThinkLite-VL-72B, on eight established VLM benchmarks. After RFT, ThinkLite-VL-7B improves the average performance of Qwen2.5-VL-7B-Instruct from 59.69% to 64.18%, and outperforms a comparable baseline trained on randomly selected 11k samples (60.89%). Similarly, ThinkLite-VL-72B raises the average accuracy of Qwen2.5-VL-72B-Instruct from 68.25% to 72.67%, exceeding the baseline trained on randomly selected 7.5k samples 69.91%.

Furthermore, compared with the most recent 7B-level reasoning VLMs, ThinkLite-VL-7B consistently demonstrates substantial performance advantages as shown in Figure 2. ThinkLite-VL-7B also outperforms much larger models—including GPT-4o, Qwen2.5-VL-72B, and o1—on the MathVista benchmark, achieving a new SoTA score of **75.1%** (Figure 1). ThinkLite-VL-72B further advances the frontier, attaining a SoTA accuracy of **79.7%** on MathVista.

**Our key contributions are:**

(1) **Difficulty as a learning signal.** We identify sample difficulty as a critical yet underutilized signal for enabling effective self-improvement in VLMs via RFT, and show the importance of scaling compute for identifying the appropriately challenging training sample.

(2) **MCTS-guided filtering.** We propose a novel use of Monte Carlo Tree Search to estimate sample difficulty by measuring model reasoning iteration count. Across diverse online and offline baselines, MCTS-guided filtering delivers superior performance, benefiting from the explicit tree search.

(3) **Data-efficient RFT pipeline.** We introduce ThinkLite-VL, a data-efficient visual reasoning framework that achieves SoTA performance using only 11k (7B) and 7.5k (72B) training samples, without any knowledge distillation.

(4) **Strong empirical gains.** We demonstrate that ThinkLite-VL-7B and ThinkLite-VL-72B outperform strong baselines and existing SoTA models across eight VLM benchmarks. Notably, ThinkLite-VL-7B improves the average performance of its base model by 7%, and achieves a new SoTA score of 75.1% on MathVista—surpassing larger models such as GPT-4o, O1 and Qwen2.5-VL-72B. ThinkLite-VL-72B further advances this with a MathVista score of 79.7%.

(5) **Open-source release.** We will release the full ThinkLite-VL model family, including both ThinkLite-VL-7B and ThinkLite-VL-72B, and MCTS-filtered training sets for both Qwen2.5-VL-7B and Qwen2.5-VL-72B to support future research in multimodal reasoning.

## 2 Related work

**Large language model reasoning.** Simulating human-like thinking processes through intermediate reasoning steps has significantly improved the performance of large language models (LLMs) on tasks that require reasoning [24]. One family of methods focuses on explicitly controlling the structure or format of the model's outputs, such as by applying Chain-of-Thought (CoT) prompting [78] and Self-Consistency [77]. Related lines of work include more elaborate reasoning strategies like Tree of Thoughts [87] or Graph of Thoughts [4]. Additionally, some approaches involve supervised fine-tuning (SFT) on curated datasets with reasoning annotations [52, 88]. Researchers have also explored process reward models (PRMs) that encourage systematic thought processes [35, 66, 70, 28, 98, 47]. Others incorporate search techniques, including Monte Carlo Tree Search (MCTS) or beam search, to refine or verify reasoning paths [80, 81, 5, 15, 19, 73]. Recently, large-scale RL with outcome-based reward functions has been leveraged [16] to elicit powerful reasoning capabilities in LLMs. Unlike prior uses of MCTS at inference time [80, 81, 15], we employ MCTS during training to assess sample difficulty and curate a high-impact training subset for RFT. We focus on how to use large-scale RL to enhance the reasoning ability of VLMs.

**Vision language model reasoning.** Vision language models [1, 68, 38, 23, 37, 3, 10, 64, 31, 85] can perform vision tasks using language given visual input through vision encoders like [57, 94, 65]. These models demonstrate comprehensive multimodal capabilities across various scenarios [93, 39, 91, 49, 18, 92, 20, 32] and exhibit reasoning capabilities to some extent [43, 76, 41, 96, 69]. Inspired by the success of reasoning in LLMs, researchers have sought to improve the reasoning capabilities of VLMs. For instance, CoT prompting is applied to VLMs [97, 51, 46, 11, 100, 21] and some papers create multimodal datasets [86, 83, 61, 99, 12, 22, 17, 63], using SFT for knowledge distillation to improve reasoning abilities. Some prior works have also explored improving VLM performance through self-improvement strategies [102, 72, 75, 13]. More recently, RL training has emerged as a promising approach to further strengthen the reasoning capabilities of VLMs [12, 22, 50, 82]. While recent works explore SFT and RL [12, 22] for VLM reasoning, efficiently utilizing training data and avoiding costly knowledge distillation remains a challenge. In contrast, ThinkLite-VL eliminates the need for SFT or distillation entirely and achieves SoTA performance using just 11k (7B) and 7.5k (72B) samples—an order of magnitude less than prior work. Specifically, we propose a novel approach using MCTS to filter for high-quality training instances based on the difficulty level. We

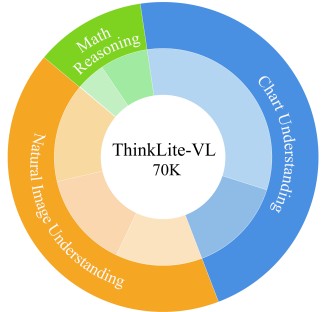

| Category | QA Category | Data source | Data size |
|----------|:-----------:|:-----------:|----------:|
| Math Reasoning | Open-ended | Geometry3K | 3001 |
| | Multi-choice | GeoQA | 5010 |
| | Multi-choice | Geos | 66 |
| Natural Image Understanding | Open-ended | FigureQA | 10000 |
| | Multi-choice | ScienceQA | 10332 |
| | Open-ended | OK-VQA | 9009 |
| Chart Understanding | Open-ended | IconQA | 10000 |
| | Open-ended | TabMWP | 22579 |

Figure 3: Data statistic of ThinkLite-VL-70k training dataset. We find that converting answers to open-ended format is critical in reliably assessing question difficulty and effective model training.

then directly apply RL training to enhance reasoning on this curated data, demonstrating strong performance without requiring any SFT stage.

**Data filtration.** Data filtration aims to identify and retain high-quality, diverse, and task-relevant data while discarding noisy or redundant information to optimize training efficiency and generalization performance. It is important for the pretraining phase [14, 29, 79, 58, 54, 2, 95, 67, 56] and instruction tuning phase [34, 33, 7, 9, 38, 103, 89] of both LLMs and VLMs. In this paper, we specifically focus on filtering training instances to curate data optimally for efficient downstream RL training to improve the reasoning capabilities of VLMs. A concurrent work, MM-Eureka [50], also investigates the impact of data filtration on RFT. While MM-Eureka [50] filters samples based on zero-shot accuracy, our MCTS-based method provides a more expressive and fine-grained estimate of sample difficulty, capturing both solved and unsolved-but-informative cases. Importantly, our findings reveal that samples requiring extended reasoning—even when not solved by the model—can be highly beneficial during RFT.

To our knowledge, ThinkLite-VL is the first framework to combine search-based sample difficulty estimation with reinforcement fine-tuning—achieving data-efficient self-improvement for visual reasoning at both 7B and 72B scale, without any SFT or distillation.

# 3 Training Recipe

In this section, we will introduce the complete training pipeline of ThinkLite-VL. First, in Section 3.1, we describe how we collect our training data that we later sample hard problems from. Then, in Section 3.2, we detail how we employ a base model combined with Monte Carlo Tree Search (MCTS) for data filtering to select prompts that are challenging for the base model. Finally, in Section 3.3, we explain how we use these filtered data to train ThinkLite-VL. We note that the proposed data filtering method, introduced in Section 3.2, is the core technical contribution of ThinkLite-VL. Specifically, ThinkLite-VL highlights the importance of difficulty-aware training sample selection in self-improving training, and effectively repurposes MCTS for sample difficulty prediction.

## 3.1 Data Collection

We collect a total of 70k datas from widely used open-source training datasets as our initial training set, covering three category: multimodel mathematical reasoning (Geometry3K [42], GeoQA [6], Geos [60]), natural image understanding (FigureQA [25], ScienceQA [43], OK-VQA [48]), and chart understanding (IconQA [45], TabMWP [44]). For FigureQA and IconQA, due to the large size of their original training sets, we only randomly sample 10k data points from each as our training set. The overall data distribution is shown in Figure 3. Each training sample is organized into the following format: (Image, id, Prompt, Answer).

Furthermore, to prevent the VLM from obtaining correct answers by merely guessing from multiple-choice options, we reformulated IconQA, FigureQA, Geometry3K, TabMWP, and OK-VQA from a multiple-choice format to an open-ended format. This modification compels the VLM to derive the

correct answer through reasoning rather than selection, thereby increasing the difficulty of the tasks and enhancing the reliability of the data filtering process described in the subsequent section.

## 3.2 MCTS-based Sample Selection

In our work, the collected data primarily originates from commonly used pretraining datasets for existing VLMs, which makes the model susceptible to overfitting on certain samples. Inspired by recent successes of data filtration in LLM SFT [53, 88] and conventional reinforcement learning [59, 74], we propose a MCTS-based sample selection mechanism. This approach leverages the VLM's own iterative reasoning process, using the number of iterations required to reach the correct answer as a metric to assess the difficulty of each data sample. Consequently, we can selectively filter for those samples that are more challenging for the model during RL training, rather than using the entire dataset.

Figure 4: Data difficulty distribution of our 11k training set after 7B MCTS-based data filtration. Unsolved refers to data that VLM cannot solve after 50 MCTS iterations.

Specifically, we define the state at step $t$, denoted as $s_t$, to represent the prefix of the reasoning chain. The introduction of a new reasoning step, $a$, transitions the state to $s_{t+1}$, which is formed by concatenating $s_t$ with $a$. By leveraging VLM itself as policy model, $\pi_\theta$, we sample candidate steps from the probability distribution $\pi_\theta(a|x, I, s_t)$, where $x$ denotes the task's input prompt and $I$ represents the input image. The MCTS process starts from the root node, $s_0$, representing the beginning of a sentence. It then iteratively proceeds through three key phases—selection, expansion and simulation—which are described in detail in the subsequent paragraphs. In contrast to previous studies, during the data filtering stage with MCTS, we prioritize computational efficiency and comprehensive exploration of the solution space, with our focus centered on self-rewarding setting. Consequently, throughout the MCTS process, we *do not employ any pretrained or separately trained process reward models*, thereby simplifying and accelerating the procedure. The prompt used for MCTS is shown in Appendix A Table 6.

**Selection.** In our MCTS procedure, the selection process is only determined by the visitation frequency, denoted as $N(s_t)$, of the current state $s_t$. At node $s_t$, the subsequent node is selected according to the following formula: $s_{t+1} = \arg\max_{s_t} \left[ c_{\text{puct}} \cdot \frac{\sqrt{N(s_t)}}{1+N(s_{t+1})} \right]$

**Expansion.** Given a current step $s_t$, the VLM generates $k$ distinct actions based on the prompt and image through temperature decoding. Each of these actions is then combined with the current step to form $k$ candidates next steps. The diversity among these actions is regulated by temperature parameter, which is set to 0.5 in our experiments, with $k$ configured as 3.

**Simulation.** After selecting a node , we directly utilize the policy $\pi_\theta$ to generate several reasoning steps until a final answer is produced or a preset reasoning step limit is reached. Subsequently, we employ the corresponding LLM (in our experiments, the Qwen2.5-VL-7B-Instruct and Qwen2.5-VL-72B-Instruct are used, with Qwen2.5-7B-Instruct serving as the critic model) to compare the generated final answer with the ground truth answer, thereby determining the correctness of the response. If the answer is correct, the MCTS process is terminated and the current iteration number $K$ is recorded; if the answer is incorrect, the visit count $N$ of the selected node is updated and the next iteration commences. Appendix A Table 7 illustrates the prompt employed for the critic model.

**Data filtration.** We apply this MCTS procedure to the entire collection of 70k data samples and record the iteration number $K$ required to solve each problem, using Qwen2.5-VL-7B-Instruct and Qwen2.5-VL-72B-Instruct as the policy model. In this process, $K$ served as a metric for assessing the difficulty of each sample: a higher $K$ indicates that the VLM requires more extensive exploration to arrive at the correct answer, thereby reflecting a greater level of challenge. Ultimately, we select all samples with $K$ greater than 5, as well as those that remained unsolved after 50 iterations, resulting in a final training set of 11k samples with 7B model and 7.5k samples with 72B model. The data difficulty distribution of 11k training set of 7B model is shown in Figure 4 as an example.

Table 1: Visual reasoning training data comparison between ThinkLite-VL-7B and other 7B-level VLM reasoning models. ALL these reasoning models have distilled knowledge from larger models or closed-source models except for MM-Eureka-Qwen-7B. MM-Eureka-Qwen-7B performs accuracy-based data filtering before training and uses more data (15k) than ours. Here the data size refers to the amount of extra visual reasoning data used to boost the base model for reasoning, via SFT or RFT.

| Reasoning Models | Knowledge Distillation (KD) | RFT | Data size |
|---|---|---|---|
| LLaVA-Cot-11B [83] | GPT-4o | ✗ | 100k |
| Mulberry-7B [86] | GPT-4o, Qwen2-VL-72B | ✗ | 260k |
| Vision-R1-7B [22] | Deepseek-R1 | ✓ | 200k + 10k |
| OpenVLThinker-7B [12] | DeepSeek-R1-Distill-Qwen-14B | ✓ | 59.2k |
| MM-EUREKA-Qwen-7B [50] | – | ✓ | 15k |
| ThinkLite-VL-7B | – | ✓ | 11k |

## 3.3 Visual Reasoning Training

Unlike previous VLM reasoning studies, which heavily depend on large-scale Chain-of-Thought (CoT) data generated by external models and employ SFT for knowledge distillation to enhance reasoning capabilities (as shown in Table 1), we demonstrate that directly performing reinforcement fine-tuning (RFT) with a small amount of high-quality training data can significantly enhance the reasoning ability of VLMs, without the need for extensive external data generation.

After conducting MCTS-based sample selection and obtaining a filtered set of high-quality training data (11k for 7B and 7.5k for 72B), we then perform RL fine-tuning on the Qwen2.5-VL models using these selected data. Specifically, we employ Group Relative Policy Optimization (GRPO) loss function proposed by [62] for training, with the objective defined as follows:

$$J_{\text{GRPO}}(\theta) = \mathbb{E}_{q \sim P(Q), \{o_i\}_{i=1}^{G} \sim \pi_{\theta}^{\text{old}}(O|q)}$$

$$\left[ \frac{1}{G} \sum_{i=1}^{G} \frac{1}{|o_i|} \sum_{t=1}^{|o_i|} \min \left\{ \frac{\pi_{\theta}(o_{i,t} \mid q, o_{i,<t})}{\pi_{\theta}^{\text{old}}(o_{i,t} \mid q, o_{i,<t})} \hat{A}_{i,t}, \text{clip}\left( \frac{\pi_{\theta}(o_{i,t} \mid q, o_{i,<t})}{\pi_{\theta}^{\text{old}}(o_{i,t} \mid q, o_{i,<t})}, 1 - \epsilon, 1 + \epsilon \right) \hat{A}_{i,t} \right\} - \beta \, D_{\text{KL}}(\pi_{\theta} \parallel \pi_{\text{pre}}) \right].$$

$$(1)$$

We provide the training prompt template during RFT in Appendix A Table 8.

## 4 Experiments

### 4.1 Benchmark Evaluation

We systematically evaluate ThinkLite-VL on several commonly used multimodal benchmark datasets and perform comprehensive comparisons with existing reasoning models. Through these experiments, we demonstrate the effectiveness and advantages of our model in multimodal reasoning tasks.

**Baseline VLMs.** We compare our method with both 7B level and 72B level models as follows:

• For 7b-level VLMs, we use Qwen2.5-VL-7B-Instruct as the base model and perform RFT on the 11k high-quality data obtained through MCTS-based filtration, resulting in our reasoning model, named **ThinkLite-VL-7B**. We conduct training using Easy-R1 [101] code base and set GRPO rollout number as 32. Our main baselines are as follows: (1) Qwen2.5-VL-7B-Instruct [3], serving as our base model; (2) ThinkLite-VL-Random11k, trained using RFT on a randomly sampled subset of 11k instances from the full 70k dataset. Besides, we report the performance of several recent general and reasoning VLMs for comparison, including general opensourced models LLaVA-Onevision-7B [30] and InternVL2.5-8B [10], the SFT-based reasoning models LLaVA-Cot-11B [83] and Mulberry-7B [86], as well as the RFT-based reasoning models Vision-R1 [22], MM-Eureka-Qwen-7B [50], and OpenVLThinker-7B [12].

• For 72B-level VLMs, we use Qwen2.5-VL-72B-Instruct as the base model. We perform RFT on the 7.5k high-quality data obtained by Qwen2.5-VL-72B-Instruct through MCTS-based filtration and get 72B reasoning model **ThinkLite-VL-72B**. The 72B-level baselines include: (1) our base model Qwen2.5-VL-72B-Instruct [3]; (2) two opensourced general VLMs LLaVA-Onevision-72B [30] and InternVL2.5-78B [10]; (3) one opensouced reasoning model QvQ-72B [71]; (4) ThinkLite-VL-Random7.5k, trained using RFT on 7.5k randomly selected samples from the full 70k dataset.

Table 2: Comparison of different VLMs on 8 widely used visual benchmarks. Our model achieves SoTA performance at both 7B level and 72B level on 6 benchmarks and reaches a SoTA performance of 79.7 on MathVista among all VLMs. On average, our model improves performance by 7.5% and 6.5% compared with our base models Qwen2.5-VL-7B-Instruct and Qwen2.5-VL-72B-Instruct. We do not evaluate Mulberry-7B on MathVision because Mulberry-7B uses MathVision as training dataset. We evaluate all models with same code using vLLM [27] inference. For reasoning models, we use thinking templates provided in their codebase to generate thoughts and get the final answer.

| Models | Data size | MathVista testmini | MathVision mini | MathVerse mini | MMMU | MMStar | MMBench | MM-Vet | AI2D | Avg. |
|---|---|---|---|---|---|---|---|---|---|---|
| Proprietary Models | | | | | | | | | | |
| OpenAI-GPT-4o | – | 63.8 | 36.8 | 50.2 | 69.1 | 64.7 | 83.4 | 69.1 | 84.6 | 65.21 |
| OpenAI-o1 | – | 73.9 | 58.2 | 57.0 | 77.6 | – | – | – | – | – |
| 7B-level General and Reasoning Vision-Language Models | | | | | | | | | | |
| LLaVA-Onevision-7B | – | 63.2 | 17.4 | 26.2 | 48.8 | 61.7 | 80.8 | 57.5 | 81.4 | 54.63 |
| InternVL2.5-8B | – | 64.4 | 22.0 | 39.5 | 54.9 | 62.8 | **82.7** | **68.8** | 83.3 | 59.80 |
| Qwen2.5-VL-7B-Instruct | – | 67.8 | 23.6 | 44.5 | 50.6 | 61.7 | 80.7 | 66.0 | 82.6 | 59.69 |
| LLaVA-Cot-11B | 100k | 54.8 | 16.3 | 33.9 | 46.2 | 57.6 | 75.0 | 60.3 | 78.7 | 52.85 |
| Mulberry-7B | 260k | 63.1 | – | 39.6 | 55.0 | 61.3 | 79.2 | 63.7 | 80.1 | – |
| Vision-R1-7B | 210k | 73.5 | 30.7 | 51.9 | 50.5 | 60.2 | 78.9 | 65.6 | 80.4 | 61.46 |
| OpenVLThinker-7B | 59.2k | 70.2 | 29.6 | 47.9 | 51.9 | 63.2 | 81.3 | 66.9 | 82.7 | 61.71 |
| MM-EUREKA-Qwen-7B | 15k | 73.0 | 31.9 | 50.3 | 52.3 | 64.1 | 79.3 | 64.9 | 81.4 | 62.15 |
| Our 7B-level Reasoning Model | | | | | | | | | | |
| ThinkLite-VL-7B-Random11k | 11k | 71.9 | 26.1 | 47.3 | 51.7 | 62.7 | 81.1 | 65.5 | 80.9 | 60.89 |
| ThinkLite-VL-7B | 11k | **75.1** | **32.9** | **52.1** | **55.5** | **65.0** | 81.4 | 67.8 | **83.6** | **64.18** |
| Δ (Ours - Random selection) | – | +3.2 | +6.8 | +4.8 | +3.8 | +2.3 | +0.3 | +2.3 | +2.7 | +3.29 |
| Δ (Ours - Open 7B SoTA) | – | +1.6 | +1.0 | +0.2 | +0.5 | +0.9 | -1.3 | -1.0 | +0.3 | +2.03 |
| 72B-level General and Reasoning Vision-Language Models | | | | | | | | | | |
| LLaVA-Onevision-72B | – | 67.5 | 29.3 | 39.1 | 56.8 | 66.1 | 85.9 | 63.7 | 85.6 | 61.75 |
| InterVL2.5-78B | – | 72.3 | 34.9 | 51.7 | 68.7 | 68.9 | 87.2 | 72.3 | **87.9** | 67.99 |
| Qwen2.5-VL-72B-Instruct | – | 74.8 | 35.2 | 53.3 | 63.4 | 68.4 | 87.4 | 76.3 | 87.2 | 68.25 |
| QvQ-72B | – | 71.4 | 32.7 | 48.6 | **70.3** | 67.2 | 86.3 | 75.9 | 86.6 | 67.37 |
| Our 72B-level Reasoning Model | | | | | | | | | | |
| ThinkLite-VL-72B-Random7.5k | 7.5k | 76.4 | 37.1 | 57.5 | 65.8 | 71.3 | 87.6 | 76.7 | 86.9 | 69.91 |
| ThinkLite-VL-72B | 7.5k | **79.7** | **43.8** | **64.3** | 68.3 | **72.0** | **88.2** | **77.3** | 87.7 | **72.67** |
| Δ (Ours - Random selection) | – | +3.3 | +6.7 | +6.8 | +2.5 | +0.7 | +0.6 | +0.6 | +0.8 | +3.06 |
| Δ (Ours - Open 72B SoTA) | – | +4.9 | +8.6 | +11.0 | -2.0 | +3.1 | +0.8 | +1.0 | -0.2 | +4.42 |

We also include proprietary models as performance references which include OpenAI-GPT-4o and OpenAI-o1. For all models, we use 8×80G A100 GPUs for model training and evaluation.

**Benchmarks.** We select eight widely used VLM benchmarks for evaluation, namely MathVista [41], MathVison [69], MathVerse [96], MMMU [93], MMStar [8], MMBench [40], MMVet [91], and AI2D [26]. Among them, MathVista, MathVison, and MathVerse are widely used in VLM research to evaluate mathematical reasoning capabilities, while MMVet also includes a significant number of mathematical reasoning tasks. In contrast, MMMU, MMStar, MMBench, and AI2D are primarily utilized to assess VLM's visual perception reasoning and scientific reasoning abilities.

**SoTA performance over both 7B and 72B models.** As shown in Table 2, ThinkLite-VL-7B and ThinkLite-VL-72B show a significant improvement in average performance across the eight benchmarks compared to the base model Qwen2.5-VL-7B-Instruct and Qwen2.5-VL-72B-Instruct, with the average performance increasing from 59.69 to 63.89 and 68.25 to 72.67, respectively. ThinkLite-VL-7B also outperforms reasoning models that primarily achieve performance enhancement through extensive knowledge distillation (such as LLaVA-CoT-11B, Mulberry-7B, Vision-R1-7B, and OpenVLThinker-7B) with the closest average performance to GPT-4o. Compared to MM-EUREKA-Qwen-7B, which does not involve SFT knowledge distillation but adopts a larger RL training dataset, our model consistently outperforms across all benchmarks, highlighting the

Table 3: Comparison with models trained on data sampled using different selection strategies, ThinkLite-VL achieves significantly better performance, highlighting the effectiveness and superiority of our proposed MCTS-based sample selection method.

| Models | Data size | MathVista testmini | MathVision mini | MathVerse mini | MMMU | MMStar | MMBench | MM-Vet | AI2D | Avg. |
|---|---|---|---|---|---|---|---|---|---|---|
| ThinkLite-VL-7B | 11k | **75.1** | **32.9** | 52.1 | **55.5** | **65.0** | 81.4 | **67.8** | **83.6** | **64.18** |
| ThinkLite-VL-Unsolved | 5.6k | 73.6 | 26.9 | 49.4 | 52.1 | 62.7 | 81.1 | 67.0 | 83.5 | 62.04 |
| ThinkLite-VL-Iter5Only | 5.4k | 73.5 | 27.5 | 50.2 | 52.5 | 64.2 | 80.9 | 66.9 | 83.3 | 62.38 |
| ThinkLite-VL-Random11k | 11k | 71.9 | 26.1 | 47.3 | 51.7 | 62.7 | 81.1 | 65.5 | 80.9 | 60.89 |
| ThinkLite-VL-SelfConsistency | 23k | 74.6 | 30.9 | 50.1 | 53.8 | 64.1 | 81.3 | 67.1 | 83.3 | 63.15 |
| ThinkLite-VL-Fullset | 70k | 74.3 | 29.9 | **52.2** | 53.1 | 63.7 | **81.6** | 67.2 | 83.0 | 63.13 |

importance of high-quality data filtering before training, and the effectiveness of the proposed MCTS-based filtering. For more discussion between offline and online data filtration, please refer to Section 4.3. Analyzing individual benchmarks, ThinkLite-VL-7B achieves best performance among all 7B-scale models on six out of eight benchmarks, with only marginal gaps behind InternVL2.5-7B on MMBench and MM-Vet. In addition, ThinkLite-VL-72B outperforms all existing open-source vision-language models across six benchmarks. Notably, ThinkLite-VL-7B attains SoTA accuracy of **75.1** on MathVista, exceeding both GPT-4o and o1. ThinkLite-VL-72B further advances the frontier, reaching **79.7** on MathVista and **64.3** on MathVerse, establishing new SoTA on both benchmarks.

**Effectiveness of MCTS-based sample selection.** Compared to training on an equal number of randomly selected samples from the full 70K dataset (ThinkLite-VL-7B-Random11k and ThinkLite-VL-72B-Random7.5k), ThinkLite-VL-7B and ThinkLite-VL-72B demonstrate a clear advantage across eight benchmarks, with average performance improvements of 5.4% at the 7B scale and 4.4% at the 72B scale. These results further show the importance of MCTS-based sample selection.

### 4.2 Importance of MCTS-based Sample Selection

We conduct ablation studies to demonstrate the importance of MCTS-based sample selection. We compare five different training settings of ThinkLite-VL: (1) ThinkLite-VL-Unsolved: Trained using only the 5.6k samples that could not be solved by MCTS, representing the most difficult subset. (2) ThinkLite-VL-Iter5Only: Trained on the subset of data that VLM is able to solve via MCTS, but required more than 5 iterations. This set, combined with the unsolved samples, forms the full 11k training set used in ThinkLite-VL. (3) ThinkLite-VL-Random11k: Trained on a randomly sampled 11k subset from the full 70k dataset, matching the size of the ThinkLite-VL training set. (4) ThinkLite-VL-SelfConsistency: Trained on 23k samples selected based on a self-consistency difficulty measure. Specifically, for each prompt, we perform 50 rollouts using Qwen2.5-VL-7B-Instruct and compute answer accuracy using Qwen2.5-7B-Instruct. Samples with accuracy lower than 0.2 are selected for RFT. (5) ThinkLite-VL-Fullset: Trained on the complete 70k dataset without any filtering. We report the evaluation results of all five settings across the eight VLM benchmarks, as shown in Table 3.

We observe that ThinkLite-VL-7B, trained using 11k samples via MCTS-guided sample selection, achieves the highest average performance among all settings. It outperforms not only the random sampling baseline but also models trained on the full dataset and self-consistency-based filtering, despite using significantly fewer training samples. This highlights the effectiveness of our difficulty-aware data selection strategy. Further analysis reveals that models trained on subsets derived solely from unsolved samples or samples requiring more than five iterations also show decent performance, suggesting that hard and medium-difficulty samples contribute meaningfully to reasoning ability. However, neither subset alone is sufficient. The combination of both unsolved and medium-difficulty samples yields the strongest and most effective training signal. Additional analyses are in Appendix B.

### 4.3 Comparison with Online Data Selection

In this section, we compare our offline data-selection strategy with an online alternative and evaluate their impact on model performance. We adopt an online baseline based on self-consistency filtering: during training we keep only those samples whose rollout accuracy is greater than 0 but below 0.9,

Table 4: Comparison between ThinkLite-VL and model trained with offline and online self-consistency based sample selection. Our method demonstrates significant advantages.

| Model Size | Training type | Selection method | MathVista testmini | MathVision mini | MathVerse mini | MMMU | MMStar | MMBench | MM-Vet | AI2D | Avg. |
|---|---|---|---|---|---|---|---|---|---|---|---|
| 7B | Offline | MCTS (Ours) | **75.1** | **32.9** | **52.1** | **55.5** | **65.0** | 81.4 | **67.8** | **83.6** | **64.18** |
|  |  | SelfConsistency | 74.6 | 30.9 | 50.1 | 53.8 | 64.1 | 81.3 | 67.1 | 83.3 | 63.15 |
|  | Online | SelfConsistency | 74.2 | 26.9 | 50.1 | 50.6 | 64.8 | **82.0** | 67.1 | 83.0 | 62.34 |
| 72B | Offline | MCTS (Ours) | **79.7** | **43.8** | **64.3** | **68.3** | **72.0** | **88.2** | **77.3** | **87.7** | **72.67** |
|  |  | SelfConsistency | 77.3 | 39.1 | 62.0 | 66.3 | 71.6 | 87.7 | 77.0 | 87.1 | 71.01 |
|  | Online | SelfConsistency | 76.9 | 38.5 | 58.2 | 66.0 | 71.7 | 87.5 | 77.1 | 87.4 | 70.12 |

drawing additional samples until the training batch is full. Table 4 compares this online variant with our MCTS-based offline selector and a plain offline self-consistency baseline. Similar to the findings in other RL studies [90], the online filter offers negligible improvement except converges faster. The decisive factor is still the ability to identify examples that are truly challenging for the current model, a task at which our MCTS selector excels due to its explicit tree search.

### 4.4 Data Difficulty Analysis between 7B and 72B Models

We analyze the 11k and 7.5k sample sets selected by 7B and 72B models, to examine how models of different capacity agree on the sample difficulty. We find that there is an overlap of 5.4k samples, where 3.6k of them are instances that neither model is able to solve within 50 MCTS iterations. The real divergence lies in the mid-difficulty stratum. We observe that for this subset, the two models often behave asymmetrically: problems easily solved by the 7B model may require many more iterations for the 72B model, and vice versa, exposing distinct reasoning heuristics across models.

We validate this model-specific preference through cross-sample training: the 11k samples selected by the 7B model are used to RFT the 72B model, and vice versa. Table 5 shows that the gains in both settings were markedly smaller than when each model trains on its own curated set. These results suggest that a sample set tailored to one model transfers poorly to another, even in a strong-to-weak setting. Instead, it is more effective to scale extra compute to find *appropriately* difficult samples that best fit the model itself, as the approach proposed in ThinkLite-VL.

Table 5: Comparison between the 7B and 72B models which trained on each other's selected samples, the resulting performance improvements drops significantly.

| Models | Data size | MathVista testmini | MathVision mini | MathVerse mini | MMMU | MMStar | MMBench | MM-Vet | AI2D | Avg. |
|---|---|---|---|---|---|---|---|---|---|---|
| ThinkLite-VL-7B | 7.5k-72B | 70.2 | 26.3 | 49.2 | 51.6 | 61.7 | 81.1 | 66.9 | 82.9 | 61.24 |
|  | 11k-7B | **75.1** | **32.9** | **52.1** | **55.5** | **65.0** | 81.4 | **67.8** | **83.6** | **64.18** |
| ThinkLite-VL-72B | 11k-7B | 76.4 | 38.5 | 58.4 | 67.2 | 70.2 | 87.3 | 76.6 | 87.4 | 70.24 |
|  | 7.5k-72B | **79.7** | **43.8** | **64.3** | **68.3** | **72.0** | **88.2** | **77.3** | **87.7** | **72.67** |

## 5 Conclusion

We have introduced an effective self-improvement approach to enhance the reasoning capabilities of VLMs, eliminating the need for external supervision or knowledge distillation. Our key insight highlights the critical importance of selecting appropriately challenging examples for RFT. We find that when training data quality is sufficiently high, even a small dataset can substantially enhance visual reasoning performance without knowledge distillation. Building on this insight, we propose a novel data selection technique, MCTS-based sample selection, which identifies and retains challenging samples by quantifying the number of MCTS reasoning iterations. Starting from 70k

initial samples, we obtain a high-quality subset comprising 11k and 7.5k challenging samples for 7B-level and 72B-level models, respectively. These curated datasets are then used to fine-tune the Qwen2.5-VL-7B-Instruct and Qwen2.5-VL-72B-Instruct via RFT, resulting in the reasoning VLMs named ThinkLite-VL-7B and ThinkLite-VL-72B. Our models demonstrate significant improvements across multiple visual reasoning benchmarks, and notably achieves a new SoTA accuracy of 79.7 on MathVista and 64.3 on MathVerse. We hope that our findings on the difficulty-based selection of RFT training data can provide insights for training more effective reasoning VLMs.

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

# A  Prompts used in experiments

## A.1  Prompt for MCTS

The prompt used for MCTS is shown in Table 6.

Table 6: Prompt used for VLM during MCTS procedure. We provide two examples of in-context learning to force VLM to follow the reasoning format.

---

**MCTS Prompt Template:**

Answer the question **step by step** and provide the final answer at the end, each step should end with **<end>** and put your final answer within ☐. Below are two examples:
Question: BoatsRUs built 7 canoes in January of this year and then each subsequent calendar month they built twice the number of canoes they had built the previous month. How many total canoes were built by BoatsRUs by the end of May of this year?
### Step1: To find the result of the total number of canoes built by BoatsRUs by the end of May, I need to find the number of canoes built in each month from January to May and then add them up. <end>
### Step2: To find the number of canoes built in each month, I need to use the formula for the number of canoes built in a given month, which is the number of canoes built in the previous month times 2. <end>
### Step3: So, the number of canoes built in January is 7, the number of canoes built in February is 7 times 2, which is 14, the number of canoes built in March is 14 times 2, which is 28, the number of canoes built in April is 28 times 2, which is 56, and the number of canoes built in May is 56 times 2, which is 112. <end>
### Step4: Now, I can add up these numbers to get the total number of canoes built by BoatsRUs by the end of May: 7 plus 14 plus 28 plus 56 plus 112, which is 217. <end>
### Final Answer: The answer is: ☐217☐.
Question: Find the number of blue circles in the figure.
### Step 1: To find the result of the number of blue circles, I need to interpret the figure. The figure is a Venn diagram with two labeled sets: - One set labeled "blue" contains all the shapes that are blue in color. - The other set labeled "circle" contains all the shapes that are circular in shape. The overlapping region of the Venn diagram contains shapes that are both blue and circular. <end>
### Step 2: The overlapping region contains shapes that meet both criteria: Blue color and Circle shape. From the diagram: - There is **one blue circle** in the overlapping region. <end>
### Final Answer: The answer is: ☐1☐.
Remember to answer the question **step by step**! Here is your question:
Question: {**QUESTION**}

---

## A.2  Prompt for Critic Model

The prompt used for critic model during MCTS is shown in Table 7.

## A.3  Prompt for RFT

The prompt used for RFT is shown in Table 8.

# B  More experiments

## B.1  Reward curves of VLM with different training data

We compare the reward curves during RFT of ThinkLite-VL-Random11k, ThinkLite-VL-Fullset, ThinkLite-VL-Iter5Only, and ThinkLite-VL, as shown in Figure 5. Although ThinkLite-VL-

Table 7: Critic prompt for MCTS simulation results evaluation.

**Critic Prompt Template:**
Please help me judge the correctness of the generated answer and the corresponding rationale.
Question: {}
Ground truth answer: {}
Generated rationale and answer: {}
Your output should only be one sentence: the generated answer is true or false.

Table 8: Prompt template used for reinforcement learning fine-tuning.

**Prompt Template:**
You FIRST think about the reasoning process as an internal monologue and then provide the final answer. The reasoning process MUST BE enclosed within <think> </think> tags. The final answer MUST BE put in ☐.

Random11k and ThinkLite-VL-Fullset achieve higher rewards during training, their actual benchmark performances are inferior to ThinkLite-VL. This observation suggests that incorporating a large number of easy samples into training rapidly improves rewards but fails to enhance the model's reasoning ability. Moreover, ThinkLite-VL exhibits notably lower rewards compared to ThinkLite-VL-Iter5Only, indicating that the unsolved data identified by our MCTS-based sample selection strategy indeed pose significant challenges to the VLM. By progressively learning to solve these challenging problems during training—even if not all are solved completely—the reasoning capabilities of VLMs can be substantially improved.

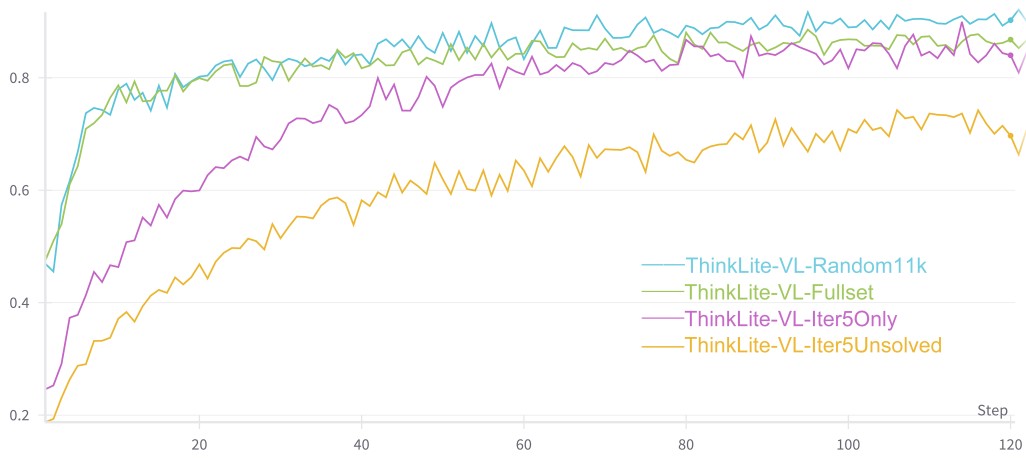

Figure 5: Comparison of reward curves of 7B models trained with different data during RFT. Iter5+Unsolved 11k dataset presents the most challenging learning setting for VLM, highlighting the difficulty of the samples selected by MCTS-based sample selection.

## B.2 Ablation Study of Data Difficulty

In this section, we investigate how training data difficulty affects model performance. We present the average performance of models trained using different difficulty data in Table 9. Notably, the model

trained with the Iter5+Unsolved subset achieves the highest average score of 63.89, outperforming all other settings. When expanding the difficulty threshold (e.g., Iter10, Iter20, Iter30, and Iter40), the model performance consistently declines, suggesting that medium-difficulty samples are important for improving model reasoning ability. As the difficulty of the training data decreases, the model's performance also declines. This trend suggests that the inclusion of an excessive number of easy samples may weaken the training signal during RFT and ultimately hurt the model's reasoning ability.

Table 9: ThinkLite-VL-7B performance under different training data difficulty settings. Iter5+Unsolved achieves the best performance.

| Difficulty level | Data size | Avg. score |
|---|---|---|
| Fullset | 70k | 63.13 |
| Iter1+Unsolved | 18k | 63.29 |
| Iter5+Unsolved | 11k | 63.89 |
| Iter10+Unsolved | 8k | 62.65 |
| Iter20+Unsolved | 6.8k | 62.61 |
| Iter30+Unsolved | 6.1k | 62.39 |
| Iter40+Unsolved | 5.8k | 62.26 |
| Unsolved | 5.6k | 62.04 |

## C  Case Studies

In this section, we present samples of varying difficulty levels selected by the MCTS-based sample selection method across different datasets, as shown in Tables 15 through 14. The difficulty levels are determined based on the number of reasoning iterations required by the VLM to arrive at the correct answer during the MCTS process, providing reference examples for understanding how the method distinguishes between easy and challenging samples.

## D  Limitations

While sample selection has effectively enhanced the reasoning capabilities of vision-language models (VLMs), the overall training efficiency remains a key limitation, both in data filtering and reinforcement learning stages. Common approaches such as self-consistency-based selection and our proposed MCTS-based strategy require substantial time for sample filtering. Additionally, the GRPO training process incurs significant computational overhead due to the large number of rollout samples needed for target value estimation. These efficiency challenges can potentially be mitigated through parallelized sample generation and processing, which we leave as an avenue for future work.

## E  Societal impacts

ThinkLite-VL can positively impact society by enabling more data-efficient and accessible development of advanced visual reasoning in vision-language models (VLMs). However, negative societal risks include the potential misuse of its MCTS-based sample selection insights for crafting sophisticated disinformation. Responsible development must also guard against over-reliance on these enhanced, yet still fallible, reasoning systems.

**Example 3: Different difficulty samples from FigureQA**

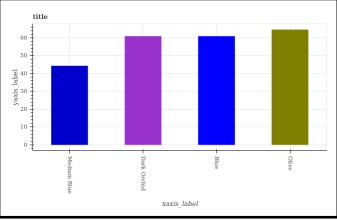

Iter0      **Question:** Is Medium Blue less than Dark Orchid?
**Ground Truth Answer**: Yes.

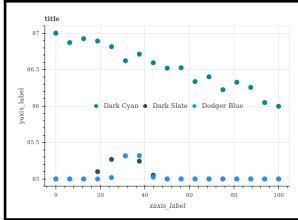

Iter29      **Question:** Does Dodger Blue intersect Dark Slate?
**Ground Truth Answer**: Yes.

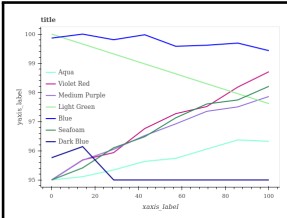

Unsolved      **Question:** Does Violet Red have the maximum area under the curve?
**Ground Truth Answer**: No.

Table 10: Example of samples with different difficulties decided by MCTS-based sample selection from FigureQA.

**Example 4: Different difficulty samples from ScienceQA**

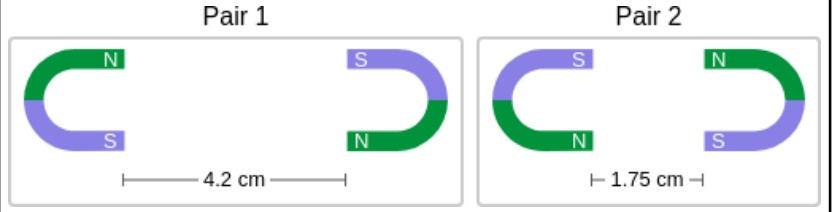

Iter0     **Question:** Think about the magnetic force between the magnets in each pair. Which of the following statements is true? Choices: (A) The magnitude of the magnetic force is greater in Pair 2. (B) The magnitude of the magnetic force is greater in Pair 1. (C) The magnitude of the magnetic force is the same in both pairs.
**Ground Truth Answer**: A.

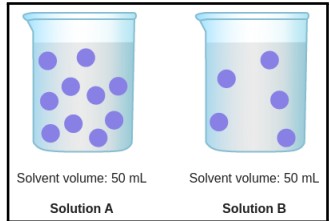

Iter13     **Question:** Which solution has a higher concentration of purple particles? Choices: (A) neither; their concentrations are the same (B) Solution A (C) Solution B
**Ground Truth Answer**: B.

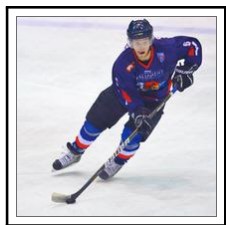

Unsolved     **Question:** What is the direction of this push? Choices: (A) away from the hockey stick (B) toward the hockey stick
**Ground Truth Answer**: A.

Table 11: Example of samples with different difficulties decided by MCTS-based sample selection from ScienceQA.

| | **Example 5: Different difficulty samples from OK-VQA** |
|---|---|

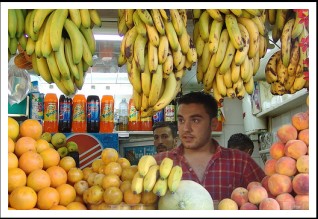

| Iter0 | **Question:** What food group is pictured here? 
 **Ground Truth Answer**: fruit. |
|---|---|

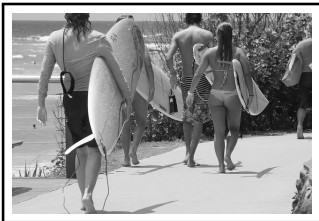

| Iter20 | **Question:** What is the length of the surfboard the man in the black shorts at the back of the line of people is holding? 
 **Ground Truth Answer**: 7 feet. |
|---|---|

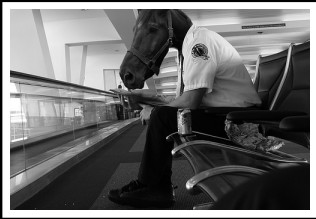

| Unsolved | **Question:** What is this guy's profession? 
 **Ground Truth Answer**: security. |
|---|---|

Table 12: Example of samples with different difficulties decided by MCTS-based sample selection from OK-VQA.

**Example 6: Different difficulty samples from IconQA**

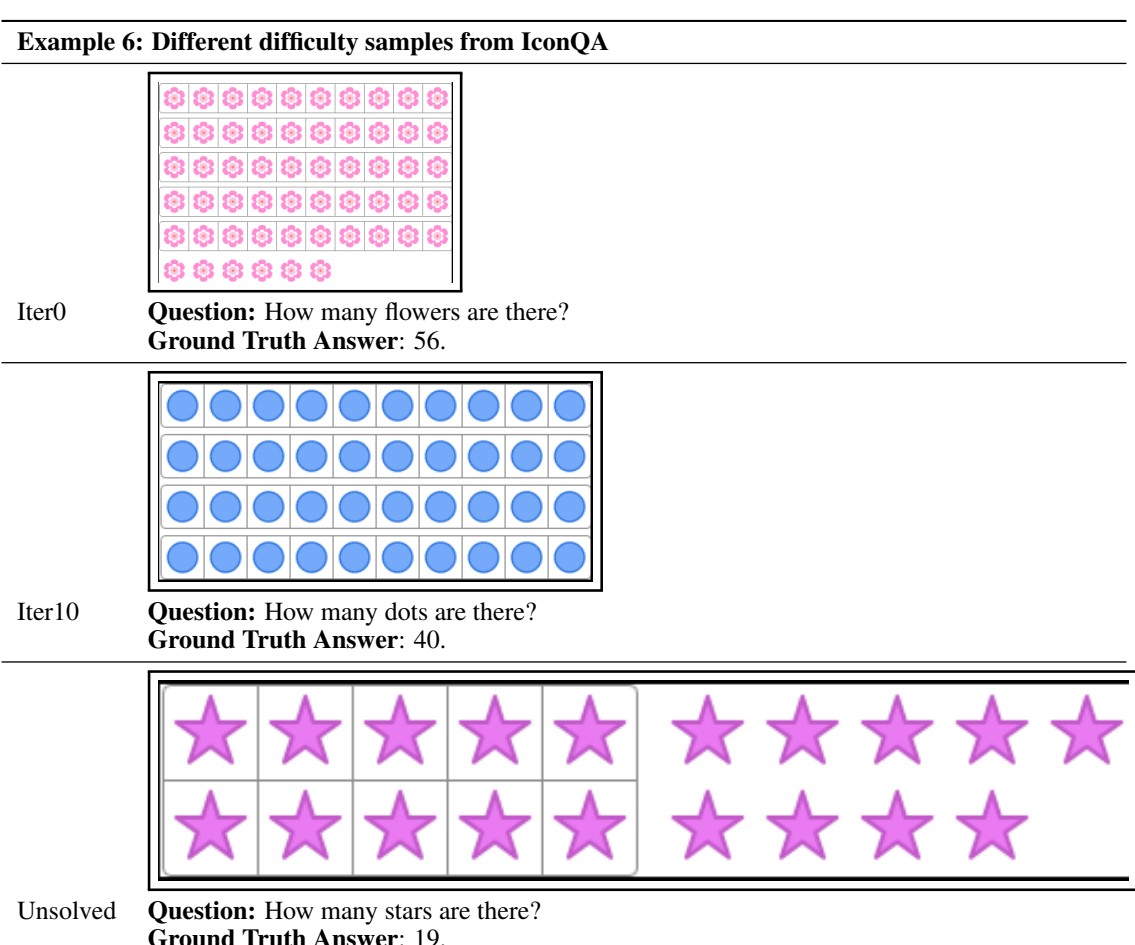

Iter0    **Question:** How many flowers are there?
**Ground Truth Answer**: 56.

Iter10    **Question:** How many dots are there?
**Ground Truth Answer**: 40.

Unsolved    **Question:** How many stars are there?
**Ground Truth Answer**: 19.

Table 13: Example of samples with different difficulties decided by MCTS-based sample selection from IconQA.

**Example 7: Different difficulty samples from TabMWP**

| | |
|---|---|
| red confetti | $11 per pound |
| gold confetti | $12 per pound |
| rainbow confetti | $10 per pound |
| silver confetti | $12 per pound |
| green confetti | $12 per pound |

Iter0    **Question:** Adriana wants to buy 3 pounds of silver confetti. How much will she spend?
**Ground Truth Answer**: 36.

| Spinning a wheel numbered 1 through 5 | |
|---|---|
| **Number spun** | **Frequency** |
| 1 | 2 |
| 2 | 9 |
| 3 | 4 |
| 4 | 11 |
| 5 | 3 |

Iter22    **Question:** A game show viewer monitors how often a wheel numbered 1 through 5 stops at each number. How many people are there in all?
**Ground Truth Answer**: 29.

**Ties per rack**

| Stem | Leaf |
|---|---|
| 3 | 2 5 6 8 9 |
| 4 | 0 4 6 8 8 8 |
| 5 | 1 4 |
| 6 | 5 8 |
| 7 | 5 6 7 9 9 |

Unsolved    **Question:** The employee at the department store counted the number of ties on each tie rack. How many racks have at least 30 ties but fewer than 70 ties?
**Ground Truth Answer**: 15.

Table 14: Example of samples with different difficulties decided by MCTS-based sample selection from TabMWP.

**Example 1: Different difficulty samples from Geometry3K**

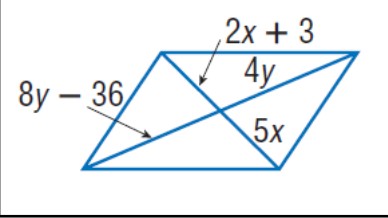

Iter0      **Question:** Find y so that the quadrilateral is a parallelogram.
**Ground Truth Answer**: 9.

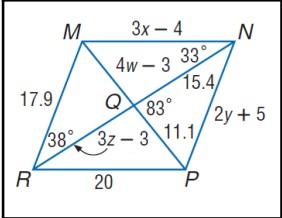

Iter16      **Question:** Use parallelogram M N P R to find y.
**Ground Truth Answer**: 6.45.

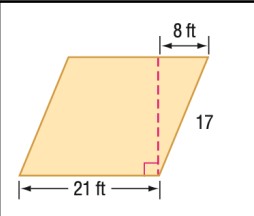

Unsolved      **Question:** Find the area of the parallelogram. Round to the nearest tenth if necessary.
**Ground Truth Answer**: 315.

Table 15: Example of samples with different difficulties decided by MCTS-based sample selection from GeoQA.

**Example 2: Different difficulty samples from Geos**

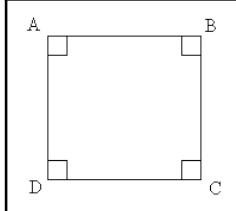

Iter0    **Question:** What is the area of the following square, if the length of BD is $2 * \sqrt{2}$? Choices: (A) 1 (B) 2 (C) 3 (D) 4 (E) 5.
**Ground Truth Answer**: D.

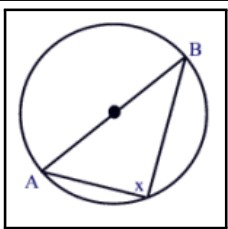

Iter7    **Question:** Given the circle at the right with diameter AB, find x. Choices: (A) 30 degrees (B) 45 degrees (C) 60 degrees (D) 90 degrees (E) None
**Ground Truth Answer**: D.

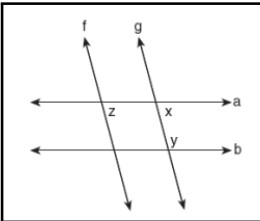

Unsolved    **Question:** In the diagram at the right, lines f and g are parallel, and lines a and b are parallel. x = 75. What is the value of y + z? Choices: (A) 75 (B) 105 (C) 150 (D) 180 (E) None
**Ground Truth Answer**: D.

Table 16: Example of samples with different difficulties decided by MCTS-based sample selection from Geos.

