# OpenReview forum: "SoTA with Less: MCTS-Guided Sample Selection for Data-Efficient Visual Reasoning Self-Improvement"
_NeurIPS.cc/2025/Conference — NeurIPS 2025 spotlight_

### Official Review · Reviewer_QP8R · 2025-06-06

**Clarity:** 3
**Significance:** 3
**Originality:** 2
**Rating:** 4
**Confidence:** 4

**Summary:**

### Summary and Contribution

- Sample difficulty is critical for effective VLM self-improvement—appropriately challenging examples that require deeper reasoning but remain solvable can drive substantial improvements even with limited data.
- The authors repurpose Monte Carlo Tree Search (MCTS) to measure sample difficulty by counting how many reasoning iterations a base VLM needs to solve each training example. This allows them to rank and filter the most optimally challenging samples.
- Starting with 70k open-source samples across math, natural images, and charts, they select only 11k samples (7B model) and 7.5k samples (72B model) based on MCTS difficulty scores, then apply pure RFT without knowledge distillation.
- ThinkLite-VL-7B achieves 75.1% on MathVista (new state-of-the-art), outperforming much larger models like GPT-4o and O1. ThinkLite-VL-72B reaches 79.7% on MathVista.

**Questions:**

- Why do you think the performance when trained on all 70K worse than in the subset of dataset?
- Can you provide a fair comparison by re-running the self-consistency baseline using exactly 11K samples (same as your method) instead of 23K? Additionally, can you clarify whether your self-consistency baseline uses chain-of-thought reasoning or direct answers, and provide results for both variants?
- How much of your performance gain comes from MCTS-guided sample selection versus the RFT training approach? Can you compare RFT vs. supervised fine-tuning on the same MCTS-selected samples to isolate these effects?
- The improvement over self-consistency (64.18% vs 63.15%) appears modest. Can you provide confidence intervals or statistical significance tests to establish whether these improvements are meaningful?
- While you claim "no SFT," you use Instruct models that have already undergone supervised fine-tuning. Can you test your approach on truly base models (non-Instruct) to better isolate the RFT contribution and validate your claims about self-improvement?
- Why does your online data selection baseline select samples with accuracy 0 to 0.9 rather than prioritizing the hardest problems (lowest accuracy)? This seems counterintuitive given your difficulty-focused approach—can you explain this choice and show results with harder problem selection?

**Ethical Concerns:**

["NO or VERY MINOR ethics concerns only"]

**Final Justification:**

I have raised my scores for quality, clarity, and significance based on the rebuttal, as the authors have addressed most of my concerns. I believe the paper will be clearer after the authors incorporate the suggested clarifications and explicitly discuss the limitations, as noted in my original review.

I have maintained my overall score, which was already more positive than negative. While the core method is not novel—difficulty-aware training with reinforcement learning has been explored in prior work—I acknowledge that this paper is submitted under the "Application" track. Within that context, I believe the authors present a well-executed application of established methods to a relevant and challenging domain.

**Limitations:**

No, their conclusion section currently does not contain any discussion of limitations. I encourage the authors to appropriately discuss the limitations, based on some of the weaknesses I highlighted above.

**Quality:**

3

**Strengths And Weaknesses:**

### Strengths

- While model performance typically scales with data size, not all researchers have access to massive datasets and compute resources. This paper demonstrates that strategic sample selection can achieve state-of-the-art results with orders of magnitude fewer training examples, making high-performance VLM training more accessible.
- The use of MCTS to quantify sample difficulty through reasoning iteration counts is creative and theoretically motivated, providing a principled alternative to random sampling or simple heuristics.
- The authors include valuable ablations comparing their approach against self-consistency methods, online data selection, random sampling, and analysis of unsolved problems, demonstrating the benefits of their MCTS-guided approach.
- Achieving new state-of-the-art on MathVista while using dramatically fewer training samples is impressive and practically significant.

### Weaknesses

While I do acknowledge multiple ablations that authors presented above, they are still lacking some key ablations. I discuss them in detail below.

- Please correct me if I am wrong, but the self-consistency baseline appears to use direct answers rather than chain-of-thought reasoning. A fair comparison should include self-consistency with CoT prompting to isolate the benefits of MCTS-guided selection.
- The self-consistency baseline in Table 3 uses 23K samples while the proposed method uses 11K. This makes direct comparison difficult—the authors should compare methods using identical sample counts (11K) to isolate the effect of selection strategy. They could either do random sampling from that 23K, or sort with increasing order of solve-rate and select first 11K. This is important, as they themselves report, training on FullSet performs worse than on carefully selected samples.
- The authors claim RFT is sufficient, but don't compare against simple supervised fine-tuning on the same MCTS-selected samples. The performance gains might stem from good data curation rather than the RL training approach.
- The improvement over self-consistency (64.18% vs 63.15%) appears modest. The authors should provide confidence intervals or significance tests to establish whether improvements are meaningful.
- While claiming "no SFT," the authors use Instruct models that have already undergone supervised fine-tuning on similar data. Testing on truly base models would better isolate the RFT contribution.
- The online data selection baseline selects samples with accuracy 0 to 0.9, but doesn't justify why harder problems (lower accuracy) weren't prioritized, which seems counterintuitive given their difficulty-focused approach.

Minor Comments:
- The paper claims both 7B and 72B models achieve "new SOTA" on MathVista, but if 72B outperforms 7B, only one can be truly state-of-the-art. This should be clarified as "SOTA within model class" or similar.
- The related work states they focus on "large-scale RL," but the core contribution is data efficiency with small-scale, carefully selected samples. This framing mismatch should be resolved.
- Lack of discussion of their limitations in conclusion section.

---

> ### Author Rebuttal · Authors · 2025-07-31
>
> We thank Reviewer QP8R for the insightful feedback. We appreciate that Reviewer QP8R found our proposed method creative and theoretically motivated, experiment results strong and extensive. Below, we address each of your concerns in detail.
>
>
>
> ---
>
> > Q1: Why do you think the performance when trained on all 70K worse than in the subset of dataset?
>
> **The 70K dataset contains many easy samples that add little learning signal.** Many examples in the full set are too easy and already solvable by the model, providing limited benefit and potentially introducing spurious patterns. By contrast, MCTS-selected samples are consistently challenging, helping the model improve reasoning rather than memorization.
>
>
>
> ---
>
> > W1&Q2: Please correct me if I am wrong, but the self-consistency baseline appears to use direct answers rather than chain-of-thought reasoning. A fair comparison should include self-consistency with CoT prompting to isolate the benefits of MCTS-guided selection.
>
> **Self-consistency baseline uses the same CoT prompt as MCTS.** We confirm that the self-consistency baseline uses identical prompting to MCTS, encouraging long-form, step-by-step chain-of-thought reasoning rather than direct answer generation. The exact prompt is provided in **Appendix A.1 Table 6**, ensuring a fair comparison between the two methods.
>
>
>
> ---
>
> > W2&Q2: The self-consistency baseline in Table 3 uses 23K samples while the proposed method uses 11K. This makes direct comparison difficult—the authors should compare methods using identical sample counts (11K) to isolate the effect of selection strategy. They could either do random sampling from that 23K, or sort with increasing order of solve-rate and select first 11K. This is important, as they themselves report, training on FullSet performs worse than on carefully selected samples.
>
> **Following your suggestion, we randomly sample 11k samples from the 23k dataset and performed Reinforcement Fine-Tuning (RFT).** The experimental results are presented in the table below. Compared to using the 23k set, the performance notably declined, indicating that self-consistency alone is insufficient to reliably identify the most challenging samples. This performance gap further underscores the effectiveness of our proposed MCTS-guided selection method in curating truly informative training data.
>
> |            | MathVista| MathVision | MathVerse | MMMU    | MMStar |MMBench | MM-Vet| AI2D |Avg.|
> | :----: |:----:|:----:|:----:|:----:|:----:|:----:|:----:|:----:|:----:|
> |SelfConsistency-11k| 73.5| 27.8| 50.6| 52.4| 63.9| 81.4| 66.9| 83.2| 62.46|
> |SelfConsistency-23k| 74.6 |30.9 |50.1| 53.8| 64.1| 81.3| 67.1| 83.3| 63.15|
> |ThinkLite-VL-7B| 75.1| 32.9| 52.1| 55.5| 65.0| 81.4| 67.8| 83.6| 64.18|
>
>
>
> ---
>
> > W3&Q3: The authors claim RFT is sufficient, but don't compare against simple supervised fine-tuning on the same MCTS-selected samples. The performance gains might stem from good data curation rather than the RL training approach.
>
> **In the experiment summarized in the table below, we perform three epochs of full-parameter Supervised Fine-Tuning (SFT) on the selected 11k hard samples, using a learning rate of 1e-5.** Compared to the RFT results, we observe a substantial drop in performance. This degradation highlights a key distinction between the two training paradigms: during RFT, the model is required to actively engage in a thinking process to derive the final answer, thereby learning to solve problems autonomously. In contrast, SFT merely encourages the model to memorize the final answers without internalizing the reasoning process, leading to poorer generalization. These findings underscore the effectiveness and necessity of RFT for enhancing the model’s reasoning capabilities.
>
> |            | MathVista| MathVision | MathVerse | MMMU    | MMStar |MMBench | MM-Vet| AI2D |Avg.|
> | :----: |:----:|:----:|:----:|:----:|:----:|:----:|:----:|:----:|:----:|
> |Qwen2.5-VL-7B-SFT-11k| 69.2| 24.2| 47.4| 50.1| 61.6| 80.6| 65.3| 82.9| 60。16|
> |ThinkLite-VL-7B      | 75.1| 32.9| 52.1| 55.5| 65.0| 81.4| 67.8| 83.6| 64.18|
>
>
>
> ---
>
> > W4&Q4: The improvement over self-consistency (64.18% vs 63.15%) appears modest. The authors should provide confidence intervals or significance tests to establish whether improvements are meaningful.
>
> We conduct a paired t-test on the evaluation results of models obtained via MCTS (Ours) and Self-Consistency at the 7B-level offline setting, as reported in Table 4 across multiple benchmarks. **The test yields a t-value of 4.28 and a p-value of 0.0027. This statistically significant difference (p < 0.01) demonstrates that the performance gains achieved by MCTS are not due to random variation**, but instead reflect a consistent and systematic improvement in overall reasoning capability.
>
>
>
> ---
>
>
> > W5&Q5: While claiming "no SFT," the authors use Instruct models that have already undergone supervised fine-tuning on similar data. Testing on truly base models would better isolate the RFT contribution.
>
> - **Since Qwen2.5-VL does not provide a base version model, we instead conduct experiments using InternVL-2.5-8B as the base model.** Specifically, we apply MCTS-guided selection to extract 19k challenging samples from a 70k dataset for Reinforcement Fine-Tuning (RFT), using the same iteration threshold of 5 as in our main setup. The training results are shown in the table below.
> - **We observe that our method significantly improves the reasoning performance of InternVL**, demonstrating strong generalization across different model architectures. Moreover, these results highlight that RFT applied to a base model can still lead to substantial gains in reasoning ability.
>
> |            | MathVista| MathVision | MathVerse | MMMU    | MMStar |MMBench | MM-Vet| AI2D |Avg.|
> | :----: |:----:|:----:|:----:|:----:|:----:|:----:|:----:|:----:|:----:|
> |InternVL-2.5-8B| 64.4| 22.0| 39.5| 54.9| 62.8| 84.6| 68.8| 84.5| 60.19|
> |InternVL-2.5-8B-ThinkLite| 69.3| 24.5| 42.9| 56.2| 64.1| 84.9| 69.5| 84.7| 62.01|
>
>
>
> ---
>
>
> > W6&Q6: The online data selection baseline selects samples with accuracy 0 to 0.9, but doesn't justify why harder problems (lower accuracy) weren't prioritized, which seems counterintuitive given their difficulty-focused approach.
>
> * **Our approach focuses on *model-centric* difficulty filtering — not curriculum learning.**
>   We assess difficulty based on the model’s own rollout behavior (via MCTS or self-consistency), and select samples that challenge the current model state. While curriculum learning imposes an ordering over data from easy to hard, our method does not enforce such a progression and is compatible with both offline and online setting.
>
> * **Exploring how to combine model-driven difficulty filtering with explicit curriculum strategies is a promising future direction.**
>   For example, one could dynamically anneal the acceptance threshold based on model improvement — but such mechanisms are outside the scope of this paper.
>
>
> ---
>
> > Lack of discussion of their limitations in conclusion section.
>
>
> - We have summarized some potential limitations of our work in Appendix D, and we will include and discuss the additional weakness you mentioned in the updated version of the paper.
>
>
>
> ---
>
> We hope these clarifications address your concerns. Thank you again for your valuable comments.

---

> > ### Comment · Reviewer_QP8R · 2025-08-05
> >
> > Thank you, Authors, for the detailed responses and for conducting additional experiments. Most of my concerns have been addressed. While I will maintain my overall rating, I have updated the other scores to reflect the improvements based on the rebuttal.
> >
> > Best of luck with your submission.

---

> ### Author Response · Authors · 2025-08-02
> **A Kind Reminder for Discussion Phase**
>
> Dear Reviewer QP8R
>
> We sincerely thank you again for your valuable feedback, which has greatly helped us improve the quality of our work. As the discussion phase is now open, we kindly remind you to review our rebuttal. We are happy to discuss more if you have any further concerns. If our responses adequately address your concerns, we kindly ask you to consider adjusting the corresponding scores. Thank you again for your time and effort!

---

> ### Author Response · Authors · 2025-08-06
> **Thanks for your response!**
>
> Thank you very much for your reply and for taking the time to review our rebuttal. We’re glad to hear that your concerns have been addressed. Two other reviewers have already updated their scores. If possible, we would greatly appreciate it if you could consider further increasing your score, as it would mean a lot to us at this stage. Once again, thank you for your thoughtful review and continued support!

---

### Official Review · Reviewer_JjLL · 2025-06-22

**Clarity:** 3
**Significance:** 2
**Originality:** 3
**Rating:** 4
**Confidence:** 4

**Summary:**

This paper presents ThinkLite-VL, which utilizes MCTS to sample data and then fine-tune through RFT. These results show that MCTS based difficult filtering can provide a scalable and effctive path thwards data-efficient self-improvement.

**Questions:**

1.	How does Figure 3 show open-ended format is critical. The caption seems not to be relevant.

2.	It seems that the proposed technique can be generally used in cross modal training. Have the authors tried this?

3.	Would the model forget old simple questions after only learning the difficult ones?

**Ethical Concerns:**

["NO or VERY MINOR ethics concerns only"]

**Final Justification:**

The reviewer is convinced by the rebuttal.

**Limitations:**

This part is relatively sound and the authors mention a meaningful limitation in the overall training efficiency. The reviewer does not penalize the corresponding limitation

**Quality:**

3

**Strengths And Weaknesses:**

## Strengths:
1.	This paper focuses the problem of sample difficulty, which might be the core problem of RFT.
2.	The proposed model seems to be generally effective across lots of benchmarks.
## Weaknesses:
1.	Although sampling data is an important topic in general LLM training, this whole paradigm looks heuristic and empirical. A concrete limitation is that this framework cannot work if MCTS cannot find the correct answer after searching, where the MCTS search may not be meaningful there. This pipeline seems to be naïve and not novel. Although the proposed data sampling might be beneficial, it is only one choice, and there could be many possibilities uncovered.
2.	About the contributions. In the paper, (3) Data-efficient and (4) Strong empirical gains seem to be a duplicate. The fifth contribution open-source release isn’t clear. The authors do not mention whether they would release the training code or only the trained model. Also, authors would better submit some code to ensure or online demos for the reviewer to assess this contribution.
3.	Writing issues:
(1)  Figure 1 and Figure 2 seem to be repetitive.
(2)  Sampled data size through MCTS should be listed in Figure 3.

---

> ### Author Rebuttal · Authors · 2025-07-31
>
> We thank Reviewer JjLL for the insightful feedback and constructive suggestions. We address each concern in detail below.
>
>
>
> ---
>
> > W1: Although sampling data is an important topic in general LLM training, this whole paradigm looks heuristic and empirical. A concrete limitation is that this framework cannot work if MCTS cannot find the correct answer after searching, where the MCTS search may not be meaningful there. This pipeline seems to be naïve and not novel. Although the proposed data sampling might be beneficial, it is only one choice, and there could be many possibilities uncovered.
>
> **We believe there may be a misunderstanding regarding our use of MCTS.** Our goal in to quantify the sample difficulty based on the number of iterations it takes for the VLM to reach the correct answer through MCTS. This difficulty metric is then used to guide training sample selection.
>
> - Specifically, we set an iteration limit for MCTS. **If the model fails to find the correct answer within the allowed iterations, the sample is marked as “unsolved” and included in the training set as a high-difficulty sample—not discarded or considered as “not working”.**
>
> - **Compared to the Self-Consistency approach, MCTS systematically explores a broader solution space, enabling the identification of more informative and challenging samples.** Our experiments demonstrate that MCTS-guided selection leads to superior model performance, highlighting its effectiveness.
>
>
>
> ---
>
> > W2: About the contributions. In the paper, (3) Data-efficient and (4) Strong empirical gains seem to be a duplicate. The fifth contribution open-source release isn’t clear. The authors do not mention whether they would release the training code or only the trained model.
>
> - **Data-efficiency refers to achieving strong performance using a small, carefully selected subset (7.5k, 11k) of reasoning data—far fewer than existing works.**
>
> - **Strong empirical gains emphasize our model's state-of-the-art results across diverse VLM reasoning benchmarks.**
>
> - **Regarding open-source commitment, we will release the full training code, training dataset, and the ThinkLite-VL model weight to support transparency, reproducibility, and future research.**
>
>
>
>
> ---
>
> > W3: Writing issues: (1) Figure 1 and Figure 2 seem to be repetitive. (2) Sampled data size through MCTS should be listed in Figure 3.
>
>
> - (1) Figures 1 and 2 serve distinct purposes. Figure 1 focuses on MathVista, showcasing state-of-the-art performance with fewer training samples, which demonstrates accuracy vs model size. Figure 2 highlights generalization **across multiple benchmarks**, emphasizing ThinkLite-VL’s consistent advantage over open-source baselines.
>
> - (2) Figure 4 presents the difficulty distribution of the final 11k hard samples. We will add sampled data size as suggested by the reviewer.
>
>
>
> ---
>
>
> > Q1: How does Figure 3 show open-ended format is critical. The caption seems not to be relevant.
>
> * **We mention the open-ended format in the caption to highlight a key preprocessing decision that enables our difficulty-aware filtering.**
>   Many source datasets originally use multiple-choice questions, which can obscure true reasoning ability (e.g., via guessing).
>   By converting them to open-ended form, we ensure that both MCTS and reward evaluation operate on free-form reasoning — which better aligns with our goal of selecting samples based on reasoning complexity rather than answer selection. We have mentioned this in **Section3.1 Line156-160**.
>
>
> ---
>
> > Q2: It seems that the proposed technique can be generally used in cross-modal training. Have the authors tried this?
>
> - **We use InternVL-2.5-8B as the base model and apply MCTS-guided selection to identify 19k challenging samples from ThinkLite-70k dataset for Reinforcement Fine-Tuning (RFT).** The MCTS sampling threshold is set to 5 iterations, consistent with our main setup. The training results are summarized in the table below.
> - **We observe that our approach substantially improves the reasoning performance of InternVL, demonstrating the generalizability of our method across different model architectures.**
>
> |            | MathVista| MathVision | MathVerse | MMMU    | MMStar |MMBench | MM-Vet| AI2D |Avg.|
> | :----: |:----:|:----:|:----:|:----:|:----:|:----:|:----:|:----:|:----:|
> |InternVL-2.5-8B| 64.4| 22.0| 39.5| 54.9| 62.8| 84.6| 68.8| 84.5| 60.19|
> |InternVL-2.5-8B-ThinkLite| 69.3| 24.5| 42.9| 56.2| 64.1| 84.9| 69.5| 84.7| 62.01|
>
>
>
> ---
>
> > Q3: Would the model forget old simple questions after only learning the difficult ones?
>
> - **Our experimental results indicate that the model does not forget how to solve simple questions after RFT.** We evaluate ThinkLite-VL on the remaining set of easy questions and find that the model still performed well. This suggests that RFT enhances reasoning ability without compromising performance on simpler tasks.
>
>
>
> ---
>
>
> We hope these clarifications address your concerns. Thank you again for your valuable comments.

---

> > ### Comment · Reviewer_JjLL · 2025-07-31
> >
> > The reviewer is in general satisfied with this rebuttal and decides to raise his score by one.

---

> > > ### Author Response · Authors · 2025-08-02
> > > **Thank you for updating your score**
> > >
> > > Dear Reviewer JjLL
> > >
> > > Thank you very much for reviewing our response and updating your score! We are glad that we were able to address your concerns, and we sincerely appreciate your valuable suggestions, which helped us further improve the quality of our paper.

---

### Official Review · Reviewer_VM7j · 2025-06-28

**Clarity:** 3
**Significance:** 3
**Originality:** 3
**Rating:** 5
**Confidence:** 3

**Summary:**

This paper introduces ThinkLite-VL, a methodology for enhancing the visual reasoning capabilities of Vision-Language Models (VLMs) through Reinforcement Fine-Tuning (RFT) using a significantly reduced number of training samples. The core contribution is a novel sample selection strategy that repurposes Monte Carlo Tree Search (MCTS) to quantify sample difficulty. This MCTS-based procedure measures difficulty by the number of reasoning iterations a base VLM (Qwen2.5-VL series) requires to correctly solve an instance. From an initial pool of 70k open-source examples, the authors filter high-quality subsets of 11k (for a 7B model) and 7.5k (for a 72B model) samples. These are then used for RFT, reportedly enabling ThinkLite-VL models to achieve state-of-the-art (SoTA) performance on several visual reasoning benchmarks, particularly MathVista, without explicit knowledge distillation during the RFT phase. The authors frame this as a data-efficient *self-improvement* process, though this improvement is critically guided by the external MCTS filtering stage.

**Questions:**

- Q1. The appendix mentions that MCTS filtering requires *substantial time*. To assess the method's overall efficiency, could you please provide a concrete breakdown of the computational cost (e.g., total GPU hours) for the MCTS filtering of the initial 70k samples, and compare this to the cost of the subsequent RFT phase?

- Q2. How sensitive is the data selection process and final model performance to this constant and other choices like the expansion temperature and the specific critic model/prompt?

- Q3. The training and test sets are largely distinct, but to rigorously test generalization, have the authors considered a stricter out-of-distribution experiment? For example, holding out all datasets from one category (e.g., *Chart Understanding*) during MCTS filtering and RFT, and then evaluating performance on benchmarks from that unseen category (e.g., TabMWP, IconQA if they have separate test splits not used elsewhere).

- Q4. Do you have recommendations on when the provided curated datasets can be effectively used versus when a full, new MCTS filtering run is necessary?

**Ethical Concerns:**

["NO or VERY MINOR ethics concerns only"]

**Final Justification:**

The authors' response effectively addresses the major criticisms and questions raised in the initial review. The previously "unquantified" computational cost is now good and shows that the MCTS filtering stage is significantly less expensive than the RFT phase, strengthening the paper's efficiency claims. The additional OOD results provide strong evidence for the generalizability of the method. While a full hyperparameter sensitivity study is still deferred, the provided insights are a reasonable first step. The authors' clarification on the reusability of the dataset and the rationale for model-specific filtering provides a clear path for future users.

**Limitations:**

Yes

**Quality:**

3

**Strengths And Weaknesses:**

$$\textbf{Strengths}$$

- The use of MCTS iterations, derived from a base model's reasoning process, to estimate sample difficulty for curating RFT training data is an intriguing approach.

- The paper demonstrates strong performance improvements using remarkably few samples (11k for 7B, 7.5k for 72B) for the RFT stage, a significant reduction compared to many existing methods that require larger datasets for similar fine-tuning efforts.

- ThinkLite-VL models achieve compelling results on eight VLM benchmarks, notably setting new SoTA scores on MathVista (75.1% for 7B, 79.7% for 72B), even outperforming some much larger models and proprietary systems.

- The success of the RFT phase without requiring knowledge distillation from larger teacher models is a notable advantage, potentially simplifying the training pipeline for reasoning enhancement.

- The paper not only finds that MCTS-derived difficulty is model-specific but also provides strong empirical justification in its appendix (Table 9) for its data selection strategy. It demonstrates that a specific mix of medium-difficulty (*Iter5+*) and very hard (*Unsolved*) samples yields optimal performance, outperforming both narrower and broader data subsets.

---

$$\textbf{Weaknesses}$$

- The paper's central claim of efficiency (*SOTA with Less*) pertains only to the number of RFT samples. The appendix reiterates that the MCTS filtering stage requires *substantial time* but provides no quantification of this cost. Knowing the GPU hours will help assess the method's true end-to-end computational efficiency.

- The appendix reveals specific prompts and fixed parameters without exploring their sensitivity. The paper does not investigate how performance is affected by key MCTS hyperparameters.

- The released datasets have limited *off-the-shelf* utility for different VLM architectures. Widespread adoption would require users to re-run the unquantified, potentially expensive MCTS filtering process for their specific models, hindering scalability.

---

> ### Author Rebuttal · Authors · 2025-07-31
>
> We thank Reviewer VM7j for the thoughtful and constructive feedback. We appreciate Reviewer VM7j finds our proposed method novel, experiment results strong and extensive. We address each of your concerns in detail below.
>
>
> ---
>
> > W1&Q1: The appendix mentions that MCTS filtering requires substantial time. To assess the method's overall efficiency, could you please provide a concrete breakdown of the computational cost (e.g., total GPU hours) for the MCTS filtering of the initial 70k samples, and compare this to the cost of the subsequent RFT phase?
>
> - **MCTS-based data selection is significantly more efficient than the RFT phase.** We report the total GPU hours required for MCTS filtering (at both 5 and 50 iterations) and for RFT with 300 training steps. We use 8xA100 GPUs for both inference and training:
>
> |            | GPU hours|
> | :----: |:----:|
> |MCTS-5 interation| 8.94|
> |MCTS-50 interation| 29.38|
> |RFT-300 step| 52.82|
>
> - **MCTS with early stopping and a small iteration cap yields practical scalability.** In our study, full MCTS runs (until a correct answer is found) were used solely to construct the difficulty distribution in Figure 4. For real deployment, we use a fixed iteration threshold K=5 (see Lines 205–207) and terminate early if the correct answer is found, greatly reducing cost without sacrificing selection quality. Moreover, our implementation of MCTS only requires searching, without the need for value function backpropagation. This significantly reduces computational complexity and makes the overall MCTS process much more lightweight and efficient.
>
> ---
>
> > W2&Q2: The appendix reveals specific prompts and fixed parameters without exploring their sensitivity. The paper does not investigate how performance is affected by key MCTS hyperparameters. How sensitive is the data selection process and final model performance to this constant and other choices like the expansion temperature and the specific critic model/prompt?
>
> **Yes, MCTS hyperparameters, especially temperature and prompt format, materially affect sample difficulty estimation and downstream performance.** While we defer a full ablation to future work due to space and compute constraints, we summarize key insights below.
>
> * **Higher expansion temperatures improve exploration and yield better difficulty signals.** Raising the temperature during expansion increases reasoning diversity, allowing MCTS to discover more plausible solution paths and making the difficulty measure more discriminative. We will include this ablation study in the final version.
>
> * **Prompt design is critical for stable step-level reasoning in MCTS.** Since MCTS operates over reasoning trajectories, prompts that enforce step-by-step structure (e.g., consistent `<end>` separators, clear final answer format) are essential. Poorly structured prompts degrade both trace quality and difficulty estimation. We provide our CoT-style prompt template in **Appendix A1 (Table 6)**.
>
> * **Critic model choice has a smaller but non-negligible effect.** We use Qwen2.5-7B-Instruct for evaluating correctness, and found that using the same model as the rollout policy ensures more consistent reward signals. We plan to benchmark against alternative critics in future work.
>
> * **We plan to include a comprehensive hyperparameter sensitivity study in follow-up work.** While our current results are robust across tested seeds and configurations, we agree that systematic ablations would further solidify the findings. We will release code and configurations to support such extensions.
>
>
> ---
>
>
> > W3: The released datasets have limited off-the-shelf utility for different VLM architectures. Widespread adoption would require users to re-run the unquantified, potentially expensive MCTS filtering process for their specific models, hindering scalability.
>
> * **We respectfully disagree — MCTS filtering is lightweight in practice, and re-executing it for different models is a necessary and principled design.** As detailed in our response to W3, filtering 70k samples with 5 MCTS iterations and early stopping requires only 8.94 GPU hours, which is ~3× faster than 50 rollouts self-consistency (27.49 GPU hours) and negligible compared to the RL training cost.
>
> * **The released ThinkLite-70K dataset remains broadly valuable and reusable.** It contains high-quality, standardized visual reasoning samples curated from diverse domains. This dataset can support evaluation, probing, or future model training even without rerunning MCTS — making it a useful resource independent of model-specific filtering.
>
> * **Reselecting samples per model is not a scalability flaw, but a feature of model-adaptive curriculum design.** Because difficulty is inherently model-dependent, applying MCTS to each target architecture ensures training samples are matched to the model’s current reasoning frontier — a key reason behind our strong results. This design parallels adaptive sampling methods in curriculum learning and RL.
>
>
>
> ---
>
> > Q3: The training and test sets are largely distinct, but to rigorously test generalization, have the authors considered a stricter out-of-distribution experiment?
>
> - **To rigorously evaluate the generalization ability of our model, we select three additional OOD benchmarks: EMMA, ZeroBench, and MMVU.**
>   - EMMA is a multimodal scientific reasoning benchmark, where all option answer candidates are images.
>   - ZeroBench focuses on natural image reasoning; we carefully verified that none of its samples appear in our 70k training dataset, ensuring a clean test for generalization.
>   - MMVU is a video reasoning benchmark that represents a fully out-of-distribution (OOD) evaluation setting.
>
> |            | EMMA mini|ZeroBench sub|MMVU val|
> | :----: |:----:|:----:|:----:|
> |Qwen-2.5-VL-7B| 24.8|13.7|45.7|
> |ThinkLite-VL-7B| 29.8|18.9|49.8|
>
> - **Our model shows consistent gains across all OOD benchmarks.**
> These results underscore the robustness and generalization capabilities of our method, even when applied to domains not seen during data selection or training.
>
> ---
>
>
> > Q4: Do you have recommendations on when the provided curated datasets can be effectively used versus when a full, new MCTS filtering run is necessary?
>
> - **Here we propose a practical strategy for applying our method to a new base model: one can first perform an initial evaluation on a randomly sampled subset from the 70k dataset.** If the model's accuracy is comparable to that of Qwen-2.5-VL, it is reasonable to directly reuse the curated 11k hard samples for fine-tuning. However, if there is a substantial performance gap, it becomes necessary to re-run MCTS-guided selection to identify hard examples tailored to the new model's capabilities.
>
> We hope these clarifications address your concerns and highlight both the efficiency and extensibility of our approach. Thank you again for your valuable comments.

---

> ### Author Response · Authors · 2025-08-02
> **A Kind Reminder for Discussion Phase**
>
> Dear Reviewer VM7j
>
> We sincerely thank you again for your valuable feedback, which has greatly helped us improve the quality of our work. As the discussion phase is now open, we kindly remind you to review our rebuttal. We are happy to discuss more if you have any further concerns. If our responses adequately address your concerns, we kindly ask you to consider adjusting the corresponding scores. Thank you again for your time and effort!

---

> > ### Comment · Reviewer_VM7j · 2025-08-05
> >
> > Authors have addressed most of my concerns and taking into account the rebuttal along with other discussions I updated the final score.

---

### Official Review · Reviewer_gSkw · 2025-07-06

**Clarity:** 2
**Significance:** 2
**Originality:** 3
**Rating:** 4
**Confidence:** 3

**Summary:**

This paper proposes a data selection method for data-efficient reinforcement learning training of visual large language models. Specifically, the authors filter training data based on offline difficulty measured by the number of MCTS iterations the model needs to solve questions correctly. Experimental results demonstrate that training models on their selected datasets achieves leading performance across multiple benchmarks on both 7B and 72B models while using significantly less training data. Ablation studies attempt to show that MCTS-based sample selection is crucial, leading to superior performance over other offline and online data selection methods.

**Questions:**

Please refer to the concerns raised in the weakness section.

**Ethical Concerns:**

["NO or VERY MINOR ethics concerns only"]

**Final Justification:**

The author's response has well addressed my concerns, the paper is clearly structured and gives several insightful findings. Therefore, I have changed my score to 4.

**Limitations:**

Yes, the authors discuss their limitations in Appendix D.

**Paper Formatting Concerns:**

No, the paper looks good in formatting.

**Quality:**

3

**Strengths And Weaknesses:**

Strengths:

- The paper is clearly written with good logical flow, clear figures, and well-designed experimental result tables that effectively communicate the methodology and findings.
- Extensive experimental validation across 8 benchmarks demonstrates that models trained on significantly less data (11k for 7B, 7.5k for 72B) achieve leading performance, showing impressive training efficiency and effectiveness of the proposed approach.

Weaknesses:

- The experimental setup for important baselines—particularly the self-consistency offline and online data selection methods—lacks clarity, raising concerns about the validity of experimental results and analysis in sections 4.2 and 4.3. Since MCTS sampling includes step-by-step reasoning before final answers, self-consistency baselines should also incorporate chain-of-thought reasoning rather than direct answers. This critical setting is not clearly specified, potentially leading to unfair comparisons.
- Several technical details about the self-consistency baselines are unclear or potentially incorrect: For offline self-consistency (line 282), samples with accuracy lower than 0.2 across 50 rollouts are selected—does "accuracy" refer to "mean accuracy@50"? Following standard practices in methods like DAPO, samples with 0 < mean accuracy@k < 1 are typically selected, suggesting the threshold should be "higher than or equal to 0.02" rather than "lower than 0.2." The rationale for this threshold selection is not provided. Similarly, for online self-consistency (line 297), the number of rollouts per sample is missing, and the upper bound of 0.9 rather than 1.0 lacks justification.
- The paper lacks discussion of inference costs for MCTS-based data sampling compared to self-consistency methods, which is crucial for evaluating practical applicability.
- Additionally, to fairly compare offline versus online data selection, the authors should evaluate MCTS-based sampling in an online setting. Without this comparison, the conclusion that "online filtering offers negligible improvement except faster convergence" cannot be generalized across all data selection methods.
- The motivation for applying this method specifically to visual large language models is unclear. The MCTS-based data sampling approach appears equally applicable to text-only model training, yet the paper lacks discussion of VLM-specific design considerations or challenges. It would be valuable to understand whether this method would achieve similar performance improvements when applied to text LLM training compared to other data selection methods, which would help establish the method's broader applicability and the specific value proposition for multimodal models.

---

> ### Author Rebuttal · Authors · 2025-07-31
>
> We thank Reviewer gSkw for the thoughtful and constructive feedback. We appreciate that Reviewer gSkw found our experiments to be well-designed, our results strong, and our proposed approach both efficient and effective in training. We address all raised concerns in detail below.
>
>
>
> ---
>
> > W1: The experimental setup for important baselines—particularly the self-consistency offline and online data selection methods—lacks clarity, raising concerns about the validity of experimental results and analysis in sections 4.2 and 4.3. Since MCTS sampling includes step-by-step reasoning before final answers, self-consistency baselines should also incorporate chain-of-thought reasoning rather than direct answers. This critical setting is not clearly specified, potentially leading to unfair comparisons.
>
> - **We clarify that both MCTS and self-consistency baselines use the same long-CoT prompting format.** For the self-consistency baselines, we use the exact same prompt as MCTS to elicit long chain-of-thought reasoning before answer generation. This ensures a fair comparison, with both methods generating step-level reasoning traces. The prompt is provided in **Appendix A1, Table 6**.
>
>
>
> ---
>
> > W2: Several technical details about the self-consistency baselines are unclear or potentially incorrect: For offline self-consistency (line 282), samples with accuracy lower than 0.2 across 50 rollouts are selected—does "accuracy" refer to "mean accuracy@50"? Following standard practices in methods like DAPO, samples with 0 < mean accuracy@k < 1 are typically selected, suggesting the threshold should be "higher than or equal to 0.02" rather than "lower than 0.2." The rationale for this threshold selection is not provided. Similarly, for online self-consistency (line 297), the number of rollouts per sample is missing, and the upper bound of 0.9 rather than 1.0 lacks justification.
>
> - **Self-consistency accuracy is defined as the proportion of correct answers among 50 long-CoT rollouts per sample.** Specifically, we compute accuracy as n/50, where n is the number of correct final answers.
>
> - **In the offline setting, we select samples with self-consistency accuracy < 0.2 as hard samples.**
>
> - **In the online setting, we select samples with accuracy in (0, 0.9) to exclude easy samples.**  We also report in the table below the experimental results based on Qwen-2.5-VL-7B when relaxing the upper bound to 1. The results show that setting the upper bound to 1 leads to inferior performance compared to 0.9, which aligns with our earlier observation in MCTS-based offline selection—including too many easy samples may undermine reasoning improvement.
>
> |            | MathVista| MathVision | MathVerse | MMMU    | MMStar |MMBench | MM-Vet| AI2D |Avg.|
> | :----: |:----:|:----:|:----:|:----:|:----:|:----:|:----:|:----:|:----:|
> |SelfConsistency (0-0.9) paper| 74.2| 26.9| 50.1| 50.6| 64.8| 82.0| 67.1| 83.0| 62.34|
> |SelfConsistency (0-1)| 73.9| 26.1| 49.3| 50.9| 64.1| 82.2| 67.0| 82.7| 62.03|
>
>
>
> ---
>
> > W3: The paper lacks discussion of inference costs for MCTS-based data sampling compared to self-consistency methods, which is crucial for evaluating practical applicability.
>
> - **We report the GPU hours required to run MCTS with 50 iterations and Self-Consistency with 50 rollouts over the full set of 70k samples, as shown in the table below.** We use 8xA100 GPUs for inference.
>
> |            | GPU hours|
> | :----: |:----:|
> |MCTS-5 interation| 8.94|
> |MCTS-50 interation| 29.38|
> |Self-Consistency-50| 27.49|
>
> - **While MCTS typically expands a much larger number of nodes compared to 50 rollout steps in Self-Consistency, in our setting, the search process is terminated early once a correct answer is found by the VLM via MCTS, and we record the actual number of iterations used.** This early-stopping mechanism significantly reduces the computational cost of MCTS, making it more efficient and feasible for large-scale sample evaluation.
>
> - Furthermore, in our study, we executed full MCTS (i.e., until a correct answer is found) across all samples solely for the purpose of analyzing the difficulty distribution shown in Figure 4. However, such exhaustive search is not required in practical applications.
> By pre-defining an MCTS iteration threshold K for sample selection (see Lines 201–208 in the paper), we can significantly reduce the computational cost. For example, in our experiments, we selected only samples with more than 5 MCTS iterations for RL training (i.e., K=5, as described in Lines 205–207). Thus, in deployment, **we can simply run MCTS with a maximum of 5 iterations, while applying the aforementioned early-stopping mechanism (i.e., terminate once a correct answer is found).** As shown in the table, **this approach reduces the total MCTS cost from 29.38 GPU hours to just 8.94 GPU hours**, which is substantially lower than the cost of Self-Consistency-50, making data selection both efficient and scalable.
>
>
>
> ---
>
>
> > W4: Additionally, to fairly compare offline versus online data selection, the authors should evaluate MCTS-based sampling in an online setting. Without this comparison, the conclusion that "online filtering offers negligible improvement except faster convergence" cannot be generalized across all data selection methods.
>
> - **We conduct both offline and online evaluations for MCTS-based and self-consistency-based selection.** The results using Qwen-2.5-VL-7B are summarized below:
>
> |            | MathVista| MathVision | MathVerse | MMMU    | MMStar |MMBench | MM-Vet| AI2D |Avg.|
> | :----: |:----:|:----:|:----:|:----:|:----:|:----:|:----:|:----:|:----:|
> |MCTS (Offline)| 75.1| 32.9| 52.1| 55.5| 65.0| 81.4| 67.8| 83.6| 64.18|
> |SelfConsistency (Offline)| 74.6| 30.9| 50.1| 53.8| 64.1| 81.3| 67.1| 83.3| 63.15|
> |MCTS (Online)| 74.7| 30.7| 51.7| 56.0| 65.2| 81.7| 67.8| 83.8| 63.95|
> |SelfConsistency (Online)| 74.2| 26.9| 50.1| 50.6| 64.8| 82.0| 67.1| 83.0| 62.34|
>
> - **MCTS outperforms self-consistency in both offline and online settings, demonstrating the robustness of our selection strategy.** Besides, offline MCTS performs best, suggesting that the offline vs. online discrepancy is an important future direction, which we plan to explore in follow-up work.
>
>
>
> ---
>
>
> > W5: The motivation for applying this method specifically to visual large language models is unclear. The MCTS-based data sampling approach appears equally applicable to text-only model training, yet the paper lacks discussion of VLM-specific design considerations or challenges. It would be valuable to understand whether this method would achieve similar performance improvements when applied to text LLM training compared to other data selection methods, which would help establish the method's broader applicability and the specific value proposition for multimodal models.
>
> * **We focus on VLMs because multimodal reasoning remains a critical open challenge, where high-quality training data and scalable self-improvement methods are still scarce.**  Unlike LLMs, which have benefited from extensive CoT-style annotations and benchmark coverage, VLMs face a **modality gap**: visual inputs are more ambiguous, grounding is harder, and reasoning requires integrating visual perception with structured logic.
>
> * **Our method addresses these VLM-specific challenges with three concrete contributions:**
>
>   * **Visual grounding under open-ended formats**: We convert multimodal QA tasks from multiple choice to open-ended form to support more faithful difficulty estimation via MCTS, creating **a curated dataset with high-quality visual reasoning samples**.
>   * **Difficulty estimation from visual reasoning trajectories**: Our approach leverages model-internal iteration dynamics (e.g., how many steps to solve a visual question), which is less meaningful in text-only settings due to more predictable language priors.
>   * **Efficient, fully self-improving visual reasoning**: Our framework demonstrates SoTA gains on 8 VLM benchmarks *without* any external distillation, directly addressing the data and supervision bottlenecks in the VLM space.
>
> * **We agree that our MCTS-based difficulty estimation is broadly applicable and plan to extend it to LLMs.**  That said, applying it effectively in the text-only domain will require new considerations (e.g., disentangling linguistic variability from genuine difficulty), which we are excited to explore in follow-up work.
>
>
> ---
>
> We hope these clarifications address your concerns. Thank you again for your valuable comments.

---

> ### Author Response · Authors · 2025-08-02
> **A Kind Reminder for Discussion Phase**
>
> Dear Reviewer gSkw
>
> We sincerely thank you again for your valuable feedback, which has greatly helped us improve the quality of our work. As the discussion phase is now open, we kindly remind you to review our rebuttal. We are happy to discuss more if you have any further concerns. If our responses adequately address your concerns, we kindly ask you to consider adjusting the corresponding scores. Thank you again for your time and effort!

---

> ### Author Response · Authors · 2025-08-06
> **Another Kind Reminder for Discussion Phase**
>
> Dear Reviewer gSkw,
>
> We sincerely thank you again for your valuable feedback, which has greatly helped us improve the quality of our work. As the discussion phase is nearing its end—with only two days remaining—we kindly remind you to take a moment to review our rebuttal.
>
> All other reviewers have already responded, expressed satisfaction with our clarifications, and accordingly updated their scores. If you have any remaining concerns, we would be very happy to engage in further discussion. Otherwise, if our responses have adequately addressed your comments, we kindly ask you to consider adjusting your score as well.
>
> Thank you once again for your time and thoughtful review!

---

### Decision · Program_Chairs · 2025-09-17

**Decision:**

Accept (spotlight)

**Comment:**

The paper proposes an approach for data selection for RFT training, based on the number of MCTS-iterations it takes the base model to solve a given training sample. The paper shows that selecting sufficiently difficult problems in this way (i.e., problems that require sufficiently many MCTS steps to solve, or are unsolvable by the base model) leads to significantly improved data efficiency during RFT, compared to other data selection methods.

All reviewers are positive about this submission, highlighting the novelty of the proposed approach and the strong empirical results for open-source models. The proposed method is simple, proven to lead to strong results, and works across multiple base models.

The reviews mentioned multiple important points that needed clarification, e.g. with regards to the computational cost of the proposed selection approach, but the authors addressed all of these during the rebuttal. One note: I would ask the authors to update their manuscript with these discussions — specifically, with the discussion about computational cost, the points about why they address VLM-specific problems, and the evaluations of their approach on other base models (InternVL). All of these strengthen the paper.

In summary, the paper is novel, easy to follow, and empirically effective. All reviewers are in favor of acceptance, and I recommend the paper as a spotlight presentation.